# Ethnobotanical Study on Plant Used by Semi-Nomad Descendants’ Community in Ouled Dabbeb—Southern Tunisia

**DOI:** 10.3390/plants10040642

**Published:** 2021-03-28

**Authors:** Olfa Karous, Imtinen Ben Haj Jilani, Zeineb Ghrabi-Gammar

**Affiliations:** 1Institut National Agronomique de Tunisie (INAT), Département Agronoime et Biotechnologie Végétale, Université de Carthage, 43 Avenue Charles Nicolle, 1082 Cité Mahrajène, Tunisia; imtinenbhj@yahoo.fr (I.B.H.J.); zghrabi@yahoo.fr (Z.G.-G.); 2Faculté des Lettres, Université de Manouba, des Arts et des Humanités de la Manouba, LR 18ES13 Biogéographie, Climatologie Appliquée et Dynamiques Environnementales (BiCADE), 2010 Manouba, Tunisia

**Keywords:** traditional knowledge, wild food, medicinal plants, ethnoveterinary, toxic, endemic, knowledge transmission

## Abstract

Thanks to its geographic location between two bioclimatic belts (arid and Saharan) and the ancestral nomadic roots of its inhabitants, the sector of Ouled Dabbeb (Southern Tunisia) represents a rich source of plant biodiversity and wide ranging of ethnobotanical knowledge. This work aims to (1) explore and compile the unique diversity of floristic and ethnobotanical information on different folk use of plants in this sector and (2) provide a novel insight into the degree of knowledge transmission between the current population and their semi-nomadic forefathers. Ethnobotanical interviews and vegetation inventories were undertaken during 2014–2019. Thirty informants aged from 27 to 84 were interviewed. The ethnobotanical study revealed that the local community of Ouled Dabbeb perceived the use of 70 plant species belonging to 59 genera from 31 families for therapeutic (83%), food (49%), domestic (15%), ethnoveterinary (12%), cosmetic (5%), and ritual purposes (3%). Moreover, they were knowledgeable about the toxicity of eight taxa. Nearly 73% of reported ethnospecies were freely gathered from the wild. The most commonly used plant parts were leaves (41%) followed by flowers and inflorescence (16%). We reported the use and collection of non-renewable parts (underground storage organs and roots) for 20 ethnospecies. Interestingly, a comparison with the available literature in Tunisia and neighboring countries reveals 13 new useful plants as well as 17 plants with new uses and demonstrates an important reservoir of traditional ethnobotanical heritage that is still sustained by respondents stemming from the semi-nomadic lifestyle of their ancestors (74% of cited taxa). These data could set a basis for further phytochemical and pharmacological research and conservative approach of the most relevant plant species including endemic overused and endangered taxa.

## 1. Introduction

Since times immemorial, wild, naturalized, or non-cultivated plants provide a “green social security” to hundreds of millions of people throughout the world, namely in the form of low-cost building materials, fuel, food supplements, herbal medicines (80% of people in the world [1]), basketry containers for storage, processing or preparation of food crops, or as a source of income [2,3].

A recent bibliometric study of all works listed in the Scopus database until 2019 (analysis of more than 100,000 publications) has underlined that global research is focused more on bioprospection rather than cultivation or domestication of plants with known biological potential [4]. This could pose a threat to the future of these natural resources. In this regard, Harshberger underlined the fact that ethnobotanical findings should not only be a record of traditional knowledge but should be relevant to current productive activities [5,6].

As an interdisciplinary science, researchers have claimed that ethnobotany is in a position to bridge various knowledge systems and policy frameworks, which could play a useful role in realizing sustainability goals that aim to alleviate poverty, ensure food security [7], provide better healthcare facilities, combat climate change, conserve biodiversity, and solve biodiversity-related issues.

Tunisian flora is one of the richest and most diverse of the southern Mediterranean with around 2700 taxa [8,9] lodging botanical species of higher interest. This flora includes 39 taxa that are endemic to Tunisia (2.3%) and constitutes a potential source of raw materials (genetic reserves, medicinal and melliferous plants, etc.). However, similar to the majority of African countries, the sustainability of these natural resources is increasingly under the high risk posed by the overuse and non-judicious exploitation especially with the globalization of medicinal plants [10].

Thanks to its geographic location, climatic diversity, and the ancestral nomadic origins of its local population, the Ouled Dabbeb’s sector in the governorate of Tataouine located in Southern Tunisia represents a rich source of plant biodiversity relevant for ethnobotanical purposes. It is a transitional zone lying between the arid and the Saharan bioclimatic belts and is characterized by low and spatio-temporal irregular rainfall of around 143 mm/year [11]. Therefore, such an environmental context has led to a remarkable richness and diversity of flora. Nevertheless, this region is still suffering from several issues related to climate change and/or anthropogenic activities that constantly affect, disrupt, weaken, and modify its ecosystems and then may further impede the close relationship developed between local populations with their natural environment that should be vested from the nomadic lifestyle of their forefathers, which relies on the surrounding biodiversity as the main vital livelihood resources.

Together with the apparent resilience and adaptive capacities to cope with such an uncertain environment, a wide-ranging knowledge is thought to be well-entrenched within the local people of Ouled Dabbeb. Such biocultural heritage deserves to be explored and documented especially given the paucity of ethnobotanical studies dealing with Tunisia’s flora [12,13,14,15,16,17], and therefore, it could play an important role in both the social and environmental sustainability of the region.

In this study, to preserve this important floristic diversity and the interesting indigenous knowledge relied on it, we had undertaken (1) botanical surveys to record and update as exhaustively as possible the list of plant taxa in this area and then identify the high-value patrimonial flora (endemic, rare, and/or threatened), (2) ethnobotanical surveys to gather data about folk uses of plants through interviews and discussions among the knowledgeable persons, and (3) comparison of reported data with the ethnobotanical literature to have a chronological view evolving plant use and have an overview on ethnobotanical transference between the current population of Ouled Dabbeb and their semi-nomadic ancestors.

## 2. Results and Discussion

### 2.1. Floristic Richness Diversity of the Studied Area

In total, 165 taxa were inventoried representing 126 genera in 48 families including 141 eudicots (86%), 22 monocots (13%), and two gymnosperms (1%) [18]. The most represented families were Asteraceae (15%), Fabaceae (10%), Poaceae (8%), Chenopodiaceae, Brassicaceae (7%), Lamiaceae (6%), Polygonaceae (4%), Apiaceae, and Caryophyllaceae (3% for each). The remaining families represented 47% of the total including 26 monospecific families.

The most predominant life forms were therophytes with 49 taxa and followed by hemicryptophytes (45 taxa), chamaephytes (42 taxa), phanerophytes (14 taxa), and geophytes (11 taxa). Nanophanerophytes were represented by three taxa and epiphytes were represented only by one taxon.

This inventory shows diversity in growth form categories (Appendix A) represented by all the growth forms with a high percentage of herbs that comprised 99 taxa (60%) followed by shrubs contributed by 33 taxa (20%), bushes having 13 taxa (8%), and shrubby trees and trees represented by 8 and 7 taxa, respectively (5% and 4%). We enumerated four taxa as subshrubs and two as parasite herbs. About 25% of the total flora recorded was composed of endemic taxa: 3 taxa are endemic to Tunisia, 14 are endemic to North Africa, 17 are endemic to the Maghreb (Northwest Africa) [9], and 6 are endemic to the northern Sahara [19].

### 2.2. Botanical Diversity of Used Plants by the Local Population of Ouled Dabbeb

Interviewees provided information about 65 ethnospecies correspondings to 70 plant species (Table 1) that belong to 59 genera from 31 families. The term “ethnospecies” used here refers to biological entities recognized by local informants, which do not necessary correspond to taxonomic biological species [20]. For example the single ethnospecies ‘Gtaf’ corresponds to three distinct taxa *Atriplex glauca* L., *Atriplex halimus* L., and *Atriplex mollis* Desf.

The most significant families in terms of number of useful taxa were Lamiaceae and Asteraceae including 8 taxa each (12%), followed by Chenopodiaceae having 5 taxa (7%) and Brassicaceae, Fabaceae, and Polygonaceae having 4 used taxa (6%) each. The 23 remaining families contributed with one to three taxa.

These results are in line with previous ethnofloristic studies conducted in Mediterranean areas that reported Asteraceae and Lamiaceae as being the ethnobotanically overrepresented families [6,21,22,23,24]. These two families are well represented in our territories in terms of number of taxa (20% of Tunisian flora estimated at 2700 taxa) with 319 Asteraceae and 183 Lamiaceae [8]. Together with the size of the family, the importance of Lamiaceae in local ethnobotany would be explained by its economic significance thanks to its aromatic plants, and particularly by the well-known important presence and diversity of its essential oil compounds [6]. Similarly, the abundance, among other compounds, of terpene compounds (including sesquiterpene lactones) in the Asteraceae taxa would explicate their prevalence to relieve different ailments, again considering the size and diversity within the family [6].

Concerning the life-form spectrum of ethnobotanical plant taxa, chamaephytes were predominant with an overall representation of 32% followed by hemicryptophytes and phanerophytes with 26% and 20%, respectively. Therophytes and geophytes were poorly represented with respective percentages of 11% and 9%. However, a previous ethnobotanical study of Le Floc’h [13] reported high use of some therophytes both in Tunisia and in North Africa. Being ephemeral taxa, available just for a short annual period [2], therophytes require an acute sense of observation as that of the nomads to note their presence in the area. So, the decrease of the utilization of these taxa recorded between 1983 and 2019 would be explained by the lack of transmission of the know-how related to these taxa held by nomadic and semi-nomadic herders to the current sedentary populations.

As was the case with their representation in the area, taxa in herbaceous habitats are used more intensively (47%) than other growth forms. This predominance can be attributed to their ubiquity (roadsides, home gardens, farmland, and in wild habitats) [25]. However, the use of this plant habit is still untenable, since it is expected that its availability decreases during the dry season and is restored only with the return of the rains [26].

We quoted the ethnobotanical use of 12 endemic (Figure 1) taxa: *Artemisia campestris* L. subsp *cinerea* Le Houér., *Teucrium alopecurus* de Noé and *Ferula tunetana* Pomel ex Batt. endemic to Tunisia, *Allium roseum* L. subsp. *odoratissimum* (Desf.) Murb. and *Henophyton deserti* Coss. & Durieu endemic to North Africa, *Searsia tripartita* (Ucria) Moffett endemic to northern Sahara, *Astragalus armatus* Willd., *Calligonum azel* Maire, *Thymus algeriensis* Boiss. & Reut., *Artemisia saharae* Pomel, *Rhanterium suaveolens* Desf., and *A. halimus*. These five latter taxa are endemic to Maghreb. Although using endemic taxa highlights the originality of traditional ethnobotanical knowledge maintained by respondents, an excessive exploration of these taxa might lead to their extinction. Thus, awareness is needed to be raised among the local population of Ouled Dabbeb focusing on the sustainable utilization and management of these plant resources.

### 2.3. Source of the Plants Used

The majority of reported ethnospecies was freely gathered from the wild (73%); in some cases (7%), they were later domesticated. However, some exotic or difficult-to-reach taxa were even cultivated (7%) or bought from herbalists (9%) on the Tataouine market. Such results revealed the interesting role of spontaneous plants in the livelihood of the local communities. These trends regarding harvesting plants from a natural environment can be linked to the nomadic lifestyle rooted in local populations of the area. Similar findings were recorded in the Algerian steppe where 60% of used plants by the nomadic community were sourced from the wild [23].

### 2.4. Revelance of Wild Used Plants in the Sector of Ouled Dabbeb

The obtained Ethnobotanicity Index (EI) in this study area was 36.96%. This means that almost 37% of the available plants in the area are considered useful by local communities of which 35% are known to be medicinal plants, 15% are known to be edible, and 4% are known to be toxic. The EI registered in our work seems to be higher than those reported in other Mediterranean studies such as 12.7% in Sicily, Italy [27], and about 27.6% in Caurel, Spain [28]. However, this ascertainment could be related to the small size of the investigated area in this study. Considering the EI of endemic taxa, the calculated value was 22.2%, which confirms the richness of local ethnobotanical knowledge maintained by the indigenous population.

### 2.5. The Parts Used and Harvest Impact

As shown in Figure 2, the most commonly used plant part was the leaves (41%) followed by flowers and inflorescence (16%) and then fruits, stems, and roots (7% for each). This prevalence of ethnobotanical use of leaves is observed in previous studies in Tunisia [17] as well as in other neighboring countries such as Algeria [21,23] and Morocco [29]. This can be explained by the fact that leaves require less effort to be obtained compared with other parts, especially underground organs, and plant exudates (gums, resins). They are also the main photosynthetic organs in the plants; therefore, they might be considered as rich sources of bioactive compounds [2]. In light of this, a recent study led by Sicari et al. [30] recommended even the consumption of blanching water of some leafy vegetable as a good source of bioactive compounds.

Since different plant resources can be grouped according to their potential for sustainable harvest, from high potential (leaves, flowers, fruits) to low potential (bark, roots, or the whole plant), information about used plant parts (Figure 2) can lead us to have a preliminary idea about yield impact and then sift out taxa that are likely considered to be more vulnerable to overharvesting [2].

In this study, debarking or the collection of non-renewable parts (underground storage organs and roots) was reported for 20 ethnospecies (17%), including two edible geophytes: *A. roseum* and *Scorzonera undulata* Vahl, which were picked up for their bulbs and fleshy roots, respectively, *F. tunetana* for its rhizome, and *S. tripartita*, *Periploca angustifolia* Labill. for their roots. It may cause enormous strain and increase mortality in taxa, thus making them vulnerable to overexploitation.

This vulnerability can be more and more alarming when it concerns endemic taxa such as *F. tunetana* and *S. tripartita.* Above all, these last two do not benefit currently from any conservation action either in situ (nature reserves) or ex situ (seed banks) [31].

### 2.6. Sociodemographic Profile of Respondents

The demographic characteristics of informants (n = 30) are summarized in Table 2. A preponderance of females (53.3%) and elders (over 60 years old) (46.6%) was noted with the majority being rural residents (80%) and illiterate (46.6%).

The number of cited ethnospecies correlated with the user’s age (Kruskal–Wallis chi-, *p*-value = 0.048) (Figure 3A) and education level (Kruskal–Wallis chi-, *p*-value = 0.046) (Figure 3B). In contrast, it was predisposed neither by gender (Kruskal–Wallis chi-, *p*-value = 0.060) nor location (Kruskal–Wallis chi-, *p*-value = 0.133). Thus, age and education are the main factors that seem to influence the local ethnobotanical knowledge of respondents. Illiterates and elderly people exhibit a more significant understanding of the ethnobotanical properties of the local flora compared to the educated and youngsters.

This can be attributed to their personal experience stemming from the semi-nomadic lifestyle of their younger years [32]. On the other hand, modernization and the exotic culture acquired by education can influence the younger generation’s unwillingness to learn. Similar results were found in another study in Morocco [21].

Although results indicated that women and men sustain a roughly similar level of ethnobotanical knowledge (Figure 4A), it was significantly patterned by the use category (F = 33.11, *p* = 2.2 × 10^−16^) and also by the interaction between gender and use categories (F = 3.0556, *p* = 0.007019).

As shown in Figure 4B, women knew more about medicinal and food taxa, whereas men were more knowledgeable about ethnoveterinary plants. Such difference can be linked to gender roles among the local community of the studied area where women are considered as primary health care and food providers for their families, and livestock keeping is the main source of income for men. Finally, it cannot be ruled out that the size of our sample (30 participants to the interviews) may have failed to disclose a highly significant variability. However, the size of the village population in itself constituted the main restraint to our sample.

### 2.7. Plant Taxa Use and Frequency

Considering that one single ethnospecies could be useful for various purposes, the results shown in Table 1 revealed that the reported ethnospecies were mainly used for therapeutic (83%) and food (49%) practices. Furthermore, informants acknowledged the use of some taxa for domestic (15%), ethnoveterinary (12%), cosmetic (5%), and ritual (3%) purposes, highlighting the wealth and wide range of ethnobotanical knowledge that is still sustained by local people.

Ten taxa were reported to be utilized for various household uses. Some of them, such as *Deverra denudata* (Viv.) Pfiesterer & Podlech, *Deverra tortuosa* DC., and *A. saharae* were used for odor control, as repellent and deterrent against tadpoles in “Magel” (a sort of Tunisian cistern made for the collection and storage of rainwater). Another interesting use concerned *S. tripartita* (root bark), *R. suaveolens* (flowers), *Olea europaea* L. (black fruit), and *Punica granatum* L. (flowers and fruit set) that seem to be used for tinting wool in light brown, yellow, pinkish white, and orange, respectively. The coloring attributes of these plant taxa were reported previously by [13] in southern Tunisia. Informants were also knowledgeable about the biopesticidal utilization of plants—namely the case of the powdered aerial part of *D. tortuosa* incorporated in the soil before sowing carrots for soil-borne pest management.

Eight taxa were used for livestock health care, intended mainly to treat skin diseases, digestive disorders, and to correct fractures. Some plants were administered to treat the same syndromes for which they are recommended in human medicine such as *R. suaveolens* and *Plantago albicans* L. that were used for their diuretic properties. Such similarity was reported in previous studies in Saudi Arabia and Palestine [33] suggesting that humans may have tried these remedies on animals before they used them for their medical problems.

*Ajuga iva* (L.) Schreb., *Lavandula coronopifolia* Poir., and *Myrtus communis* L. are three aromatic taxa that were widely used by women for cosmetic purposes, mostly for hair care and to concoct perfume commonly homemade for the brides in this region. The uses of these three species were also reported by Ghrabi-Gammar et al. [34].

The interviewees seldom referred to religious and ritual customs. Only two ethnospecies were reported for this type of use: the fruits of *Ziziphus lotus* Lam. were used for washing and shrouding the deceased thanks to their saponifying properties, and *Ruta chalepensis* L. was used to ward off the evil eye and spirits by inhalation of burned leaves. The same practice was reported by Le Floc’h [13] in Tunisia.

#### 2.7.1. Relative Frequency of Citation (RFC)

The RFC of the reported taxa ranged from 0.03 to 1 (Table 1). The ten highest values of RFC corresponded to the following taxa: *Haloxylon scoparium* Pomel (RFC = 1), *Moricandia arvensis* (L.) DC. (RFC = 0.73), *A. campestris* (RFC = 0.77), *Retama raetam* (Forssk.) Webb & Berthel. (RFC = 0.67)*, Capparis spinosa* L. (RFC = 0.60), *A. saharae* (RFC = 0.70), *A. roseum* (RFC = 0.53), *Citrullus colocynthis* (L.) Schrad. (RFC = 0.53), *Rosmarinus eriocalyx* Jord. & Fourr. (RFC = 0.53), and *S. tripartita* (RFC = 0.60). These different plants were cited by at least 13 informants out of 30 (almost half of total respondents), representing the most common and solicited useful plants by the local population. These findings demonstrated the perception and widespread use of these taxa by local inhabitants, which make them well recommended for further pharmacological studies.

#### 2.7.2. Human Medicine

Our surveys stated the use of 54 ethnospecies for various therapeutic purposes that fall into 17 ailment categories (Table 3). Among them, 22 taxa (40.7%) were devoted to treat digestive disorders (hemorrhoids, constipation, diarrhea, ulcers, bloating) with 72 citations. Eleven taxa (20.4%) were used to relieve dermatological problems (wounds, irritations, dermatoses, furunculosis, eczema, spots) with 49 citations, while nine taxa (16.7%) were revealed as remedies for cold, respiratory diseases (bronchitis, cough, asthma, sore throat) and fever with 42 citations. To treat either urogenital disorders (urinary incontinence) or cardiovascular disorders, the local population described the use of 10 taxa (18.5%) for each ailment category. Six taxa (11.1%) were mentioned as remedies for each case of headaches, endocrine system disease (diabetes), and sexual reproductive problems while only five taxa (9.3%) were used to relieve muscular/skeletal diseases (rheumatism, arthritis, and joint pain). Four taxa (7.4%) were reported for treating fractures, toothache, and mouth inflammation. Two ethnospecies (3.7%) were used to soothe nervous system diseases as a tonic or as calming and sedative taxa, whereas two other ones were used for the ear–nose and throat (ENT) conditions. Finally, the interviewees stated the use employment of one taxon for eye care and another for cancer treatment. In line with these observations, previous studies underlined that the local people of the northwest of the governorate of Kef in Tunisia [17] and also nomadic peoples in the Algerian steppe [23] were mainly informed about plants that relieve digestive system disorders, skin diseases, and respiratory system diseases.

##### Informant Consensus Factor (ICF)

The Informant Consensus Factor (ICF) was calculated for all ailments categories, except ophthalmological diseases, which had only one citation (Table 3) and was found to range from 0.40 to 1.00 with an average of 0.68. Such results demonstrate that knowledge about the therapeutic use of plants seems to be well distributed among the local population of the area of the study.

Cancer had the highest ICF value (ICF = 1.00) due to the use of a single species for its treatment (*M. arvensis*), demonstrating that local pharmacopoeia could provide species with promising anticancer activities. In this context, a pharmacological study [35] evaluated this species and highlighted the antiproliferative activity of its leaf extract against human cancer cells.

Poisonous animal bites presented the second-highest ICF value (ICF = 0.82) and were reported to be treated using three taxa *A. campestris*, *C. colocynthis*, and *R. raetam*. This may be attributed to frequent reports about scorpions and snakes in Saharan areas or to the paucity of alternative treatments and the long distancing from hospital centers. Similar results were also observed in southern Tunisia [13] and among the nomadic populations in Algerian Sahara [36].

Considerable consensus values were also found for the treatment of dermatological illnesses and bone pains (ICF = 0.79) for each. Other categories were noted with high ICF values such as fever (ICF = 0.75), sexual reproductive problems (ICF = 0.71), and gastrointestinal disorders (ICF = 0.70).

Less agreement was observed in treating kidney urinal tract problems (ICF = 0.53), toothache, and mouth inflammation (ICF = 0.40).

##### Preparation and Modes of Administration

In our study, decoction was the most commonly applied method of preparation (43% of total used medicinal taxa) followed by infusion (16%), powder (11%), poultice, and maceration (10% for each) (Figure 5). Similar findings were reported by other studies [21,37]. Regarding direct usage, five cases (6%) were noted in which fresh plant parts were chewed mostly to relieve gastrointestinal ailments. For example, respondents disclosed that tubercles of *Erodium glaucophyllum* (L.) L’Hér. were directly eaten for their laxative properties.

Once plants have been prepared, the next step is to proceed with the administration. The two main roots of administration recorded in the present work were oral application (64%) and topical applications (36%). Most of the plants were used as they are. However, in some cases, they were mixed with adjuvants such as milk and olive oil to potentiate their effects and facilitate their application. For instance, a decoction in the milk of leaves powder of *Nitraria retusa* Asch. was used to prepare an ointment to treat leishmaniasis. Few species were used in combination with other plants. Particularly, aerial parts of *Cynodon dactylon* (L.) Pers. were boiled with leaves of *Polygonum equisetiforme* Sm. to treat urinary incontinence.

#### 2.7.3. Culinary Uses

Our survey highlights that 32 ethnospecies were reported as being traditionally used for culinary purposes (Table 1). Slightly more than their quarter (26%) are consumed raw. Some are used in salads such as the tubercles of *E. glaucophyllum* (Tommir) and leaves and flowers of *Rumex roseus* L. (Homidha). The latter are appreciated for their lemony taste. Others are often eaten directly off the plant as a snack. As examples, we can cite the fruits of jujube *Z. lotus* (Nbeg) and *S. undulata* (*Guiz*), which is a plant consumed for its fleshy root and flowers, which taste similar to chocolate. Nevertheless, the majority of edible reported ethnospecies are consumed freshly cooked as a part of typical savory dishes, such as couscous, vegetable stews, and thick soups. The leaves and bulbs of the two Alliaceae: *A. roseum* subsp. *odoratissimum* and *Allium ampeloprasum* L. are often incorporated in the traditional flatbread ‘Kesra’. *Asphodelus tenuifolius* Cav. leaves are freshly cooked as an ingredient of the sauce for couscous and also to prepare the traditional soup ‘Dchich’. Similar findings were reported in Sidi Bouzid (central Tunisia) where a favorable perception of these same wild edible plants was observed [38]. This could spur their consumption at a national scale as a part of sustainable food systems for good nutrition and health, as several studies cited by the Biodiversity International revealed that the nutritional value of wild and indigenous food is often higher than that of their cultivate counterparts [39].

In addition, we reported the use of some plant taxa either as condiments such as *T. algeriensis* and *R. eriocalyx* leaves or as flavoring mainly added to black tea, as is the case of twigs of *A sahareae* and bark of the roots of *P. angustifolia*. We recorded also some “cognitive use” and “not-so-active use” regarding some ethnospecies used in times of food insecurity in the past, in the 1940s, as a replacement or complement to barley flour namely grounded dried leaves of *A. halimus*. This was confirmed by Bouquet’s study [40] focusing on food in Tunisia at the beginning of the 20th century.

Several reported edible plant taxa are considered as functional foods that are ingested in a food context as culinary preparations for their assumed health benefits [41]. Powders of *F. tunetana* (rhizome) and *S. tripartita* (peeled roots) are added to barley flour to prepare a traditional dish locally named ‘Ich’ for its mineralizing and anti-inflammatory proprieties as well as to relieve ulcers. Likewise, leaves and flowers of *M. arvensis* are commonly consumed by local people of Tataouine during spring in thick soup and are perceived as a panacea that boosts the immune system and may treat or prevent the development of cancer.

#### 2.7.4. Toxicity of Used Plants

Eight taxa were reported as being toxic for humans and/or livestock. As shown in Table 1, their poisonous effects were manifested by gastrointestinal disorders, skin irritation, and abortifacient effect, and they depended mainly on the consumed quantity. Moreover, serious poisoning cases inducing sometimes death were reported after oral administration of *R. raetam* and *C. colocynthis*. In this regard, previous pharmacological studies confirmed the lethal effect of the pulp extract of *C. colocynthis* received by rabbits [42] and also its toxicity on liver cells which may engender hepatocyte necrosis and liver fibrosis [43]. Likewise, the findings of Algandaby [44] revealed that the repeated administration of excessive doses of methanolic extract of *R. raetam* in rats could exhibit hepatotoxic, nephrotoxic, and mutagenic effects.

An overlap of five taxa was found between toxic plants reported in this investigation and those referenced by Le Floc’h [13] (Figure 6). Additionally, nine taxa were mentioned in the literature to be poisonous while they were frequently utilized by local communities of Ouled Dabbeb. Four of them were cautiously used for the control of snakebites, insect pests, and many other harmful organisms.

Some experimental studies proved the involvement of endogenous alkaloids of *Peganum harmala* L. in Parkinson’s disease [45] and the potent tetratogenic effect of *Nicotiana glauca* Graham high in anabasine [46] confirming, thus, their poisonous effect described by Le Floc’h [13]. The ignorance of such possible toxicity may interfere with their biosafety, since the mode of application described by respondents seems to be “unsafe” (Figure 6). Nevertheless, earlier studies proved positively the innocuousness of *Diplotaxis harra* Boiss., *A. tenuifolius* [47,48], and *C. dactylon* [49,50]. Therefore, the complexity of issues drives the need for further research of the phytochemical analyses and toxicological effects of the documented flora to provide a better understanding about the safety and efficacy of their consumption.

### 2.8. Bibliographical Comparison

#### 2.8.1. Transmission of Ethnobotanical Knowledge over Time (1936–2016)

The comparison of reported data with the ethnobotanical literature revealed that local plant knowledge in the study area may have changed, been lost, or newly developed, but also elements of this knowledge may persist for many generations (Figure 7). We note that the interviewees in this study ignore the use of over 38 taxa available in their surrounding area while they have been widely used by their nomadic and semi-nomadic forefathers in the past (Appendix A). Among these plants, we find mainly those of critical survival value namely edible ones used in times of food insecurity such as *Aizoon canariense* L. and *Erodium crassifolium* (Forssk.) L’Hér. [51]. We also assess taxa with detergent proprieties such as *Anabasis articulata* (Forssk.) Moq. and *Fagonia cretica* L. Knowledge on medicinal plants also seems to have reduced especially that on skin healing wound flora as *Plantago lanceolata* L., *Limoniastrum guyonianum* Durieu ex Boiss., and *Thymelaea microphylla* Coss. & Durieu.

Nonetheless, this comparison demonstrated that an important reservoir of traditional ethnobotanical heritage among respondents is still present. More than 74% of cited taxa in this study were also perceived as usable by nomads and semi-nomads in the past. It should be noted that despite decades in decline, local inhabitants of Ouled Dabbeb sustain the same use of 29 taxa, keeping the same method of preparation and administration adopted by their nomadic ancestors. This shared knowledge essentially concerns edible (35%), medicinal (35%), and toxic (17%) plants (Figure 7e’).

The data presented here mirror the above-reported wealth and experience of nature gained over generations. Thus, we can talk about indigenous ethnobotanical knowledge, which is defined by Masango as “the totality of all knowledge and practices established on past experiences and observations that is held and used by people”, which merits further valorizations.

In addition, some preparations recorded by Le Floc’h [13] were newly developed by the current inhabitants of Ouled Dabbeb in order to meet their present-day needs.

For some taxa such as *Haloxylon schmittianum* Pomel, *P. albicans*, and *R. chalepensis*, the same preparation used in the past for human health care are cited in this study to treat animal diseases. Likewise, for some taxa such as *P. granatum* L. and *Malva aegyptia* L., participants merge medicinal knowledge inherited from their immediate ancestor with dietetic knowledge recently acquired to prepare some functional food. These results suggest that some past ethnobootanical knowledge is transmitted from parent to offspring and subsequently expanded upon and incorporated to include a broader pattern of use.

#### 2.8.2. New Plant Reports and Uses

A comparison of reported data with the ethnobotanical literature revealed that 48 of the used taxa in the current study were reported as useful in other regions of Tunisia as well in other neighboring countries (Algeria and Libya). We quoted a considerable similarity concerning plant use and mode of application (34 taxa). Shared plants are mainly cosmopolitan accessible taxa well known in the Mediterranean areas. Compared to ethnobotanical surveys carried out in Sahara Septentrional [52] and among nomadic and Touareg people in the Algerian steppe [23,36], the ethnobotanical knowledge recorded in this study area has an important similarity level with that quoted in Algerian regions. This overlap could be related to the similar Saharan environment in these two regions and also to the similar cultural ancestry that they shared. However, in comparison with Lybia [53], we recorded a huge difference regarding patterns of use and application despite its enormous socio-cultural, geographical, lifestyle, and ethnic similarity with Tunisia. Such a difference could reflect the lack of horizontal transmission of ethnobotanical knowledge between the two neighboring areas.

Finally, this comparative analysis led us to identify 11 new useful plants as well as 16 plants with new uses (marked with asterisks in Table 1) that have not been recorded previously in compared ethnobotanical bibliography. This mainly included endemic taxa such as *F. tunetana* and *R. eriocalyx* taxa, whose uses are suggested to be related to the challenges of the 21st century (cancer, stress, pesticide pollution) namely *D. tortuosa* for its biopesticidal properties and *M. arvensis* for treating and preventing the development of cancer. This study recorded for the first time the folk use of *N. retusa* to relieve mucocutaneous leishmaniasis vector-borne disease, which is a vector-borne disease that is highly endemic in Tunisia [54] and particularly in Ouled Dabbeb, where the number of cases recorded has multiplied by at least 20 times since 2011 according to the last study carried out in 2012–2013 [55]. Thus, this taxon may be a promising source for the ongoing research for plant-derived leishmanicidal compound conducted at a national scale [56,57], especially given the high toxicity of the available drug [57]. The data observed here highlighted the originality and uniqueness of the relevant ethnobotanical knowledge sustained by the local population of the Ouled Dabbeb Sector.

## 3. Material and Methods

### 3.1. Study Area

The sector of Ouled Dabbeb (32°53′03′′ N and 10°16′46′′ E; 32°53′03′′ N and 10°28′55′′ E; 32°43′54′′ N and 10°16′46′′ E; 32°43′54′′ N and 10°28′55′′ E) is located in the south-east of the goverenorate of Tataouine in the south of Tunisia and covers an estimated area of 100 km^2^. It corresponds to a vast intermountain depression framed by two cuestas characteristic of the Plateau of Matmata (Figure 8). This depression is drained by the occasional flow of some streams with the wadi Tataouine and the wadi Jerjer being the main ones. The highest point of this area belongs to the Jebel Bou Louha, in the west of the village of Ouled Dabbeb with an altitude of 662 m.

Enclaved between two Mediterranean bioclimatic domains—the arid-lower bioclimatic stage in its northern, western, and southwestern parts and the upper saharan stages in the southern and southeastern part—the sector of Ouled Dabbeb is characterized by low and spatio-temporal irregular rainfall of around 143 mm/year and a prolonged dry season that extends from April to September. Temperatures can reach a maximum of 49 °C in Aout, while the mean minimum temperature is recorded in December, January, and February (15.5 °C) [11]. This contrasting thermal regime with potential evapotranspiration (peak of 170 mm in July) leads to water deficits that reach 2106 mm [11].

According to the latest census (2014), the number of the population in this study area is 49.137, of which 88.8% are in rural areas.

From the late twentieth century until recently, the region was exploited by nomadic and semi-nomadic herders practicing seasonal livestock movements at the local, regional, and national scale [58]. Villages and houses served as storage for families’ grain and wealth and as a meeting point for social events [59]. Nowadays, with the sedentarization of the population and the privatization policy of the rangelands, the steppe area decreased in favor of cropping and arboriculture. A field survey carried out in 2014 by Abaza and Hanfi [11] shows that the region’s economy of the region is based mainly on subsistence agriculture devoted to sporadic cereal crop production and rustic arboriculture (olive and fig) and livestock breeding (sheep, goats, and camels).

Although these agropastoral activities reflect a relative adaptation to the environmental conditions, it does not hide the great precariousness of this region, which is notably due to inappropriate use of pasture, which could pose a threat to its biodiversity. For the breeding of livestock, it is desirable to provide a grazing plan to preserve natural habitats [60]. For this reason, the application of some management practices, such as adopting (1) a moderate grazing program that is well known as a tool of managing floristic biodiversity [61], (2) livestock’s breeding strategies in family projects well used in arid regions in Tunisia [62], and (3) the deferred grazing, becomes a necessity for optimizing ecosystems’ productivity and conserving biodiversity.

Despite ecological harshness (long dry season, soil erosion, wind deflation, strong steady agropastoral activities...), the apparent monotony of the sparse sub-Saharan steppe vegetation in this area hides very rich and diversified flora including rare taxa [63,64] with a great heterogeneity of the landscape and vegetation in a small territory. According to previous studies [11,63], four main plant groupings were outlined: (1) *Macrochloa tenacissima* (L.) Kunth and *Artemisia herba-alba* Asso grouping that dominated the surrounding mountains, (2) *A. herba-alba* and *H. scoparium* grouping characterized loamy arid piles of the study area, (3) *R. suaveolens* and *A. campestris* grouping that defined the sandy plains, and (4) *Thymus capitatus* Hoffmanns. & Link and *A. campestris* grouping that marked the wadi beds.

### 3.2. Ethnobotanical and Floristic Surveys

Three ethnobotanical surveys were conducted, each lasting at least six days during May 2014, January 2015, and March 2016 in the district of Tataouine covering mostly all seasons of the year. Folk uses of plants were gathered through interviews and discussions among the knowledgeable persons (lay practitioners, traditional herb sellers, shepherds, and village elders). To ensure a comprehensive representation of the local knowledge and use of plants, the choice of the respondents was heterogeneous in terms of gender, age, and level of education. A total of 30 informants (16 women and 14 men) were interviewed. Their age ranged from 27 to 84 years. Most of the respondents claimed they were born in the region and had lived there all their life.

Researchers adhered to the Code of Ethics of the International Society of Ethnobiology [65] and always explained the purpose of the study and obtained verbal informed consent before conducting interviews.

Surveys were carried out in local inhabitants’ houses, in fields, and in herbalists’ shops, where informants were requested to show the researchers the taxa used for diverse cited purposes. All interviews were carried out in the Tunisian dialect.

The questionnaire comprised two sections: The first compiled the main demographic and social features of the participants (gender, age, education rate, occupation, etc.), and the second gathered information on local communities’ folk knowledge about the plant taxa growing in their surrounding areas namely vernacular names of used taxa, uses (medicinal, food, ethnoveterinary, domestic, ritual, etc.), used plant part(s), methods of preparation, administration procedures (in the case of medical remedies), growth form [16,66,67], and life form [16,19,66,67].

Within the scope of this study, reasonably complete flora surveys were undertaken from May 2014 to April 2019 exploring all sides of the studied territory including desert, mountain plains, and river valleys to compile an exhaustive list of all vegetation types within the study area. Such a complete list constituted a baseline to well estimate the importance of useful plants in this area.

### 3.3. Botanical Identification

Samples of plant taxa either recorded during floristic surveys or mentioned by our informants were collected at the flowering/fruiting stages for the accurate identification and preparation of herbarium specimens. In addition, photos were taken in situ when available. Plant materials were preserved using standard procedures and identified by the authors according to Flora of Tunisia [12,66,67], Flora of North Africa [68], and flora of the Sahara [19]. Taxonomic identification was updated concerning systematics nomenclature and chorology using Commented Synonymic Catalogue of the Flora of Tunisia [8], and the index of the database of North Africa [9]. The herbarium specimens were coded and deposited in the herbarium of the National Agronomic Institute of Tunisia (INAT) [69]. Instances of endemism and hazard categories [9,66,67] were specified as per information availability.

### 3.4. Statistical Analyses and Quantitative Indices

To describe the gathered ethnobotanical information, data were first tabulated into an Excel database and structured in Use Reports (UR, a citation of one plant use by one informant). Then, it was analyzed and assessed using various quantitative indices, including, the Relative Frequency of Citation (RFC), the Informant Consensus Factor (ICF), and the Ethnobotanicity Index (EI).

A bibliographical comparison with ethnobotanical studies conducted in Tunisia and neighboring countries (Algeria, Libya) was performed in order to evaluate the originality and uniqueness of ethnobotanical knowledge sustained by the local population of Ouled Dabbeb and to identify new plant citation and new uses. We collected data available from academic databases including PubMed, Springer, Science Direct, and Scopus, using keywords or combinations of keywords linked to the context of this study: Tunisia, Algeria, Libya, Sahara, ethnobotany, steppe, folk uses of plants, and nomads. Furthermore, [13,14,15] were used to compare fieldwork results with historical texts, as these sources are considered to be the first pioneering works on ethnobotany in Tunisia that provide an interesting background, traditional botanical knowledge present in various Tunisian regions, and more specifically in the southern part of the country. The authors of these studies adopted a documentary analysis as secondary data to attain more complete coverage of historical aspects on the topics covered in their ethnobotanical surveys. They used manuscripts, letters, old books, and documents from government and related organizations that date back to the end of the 19th century.

This comparison can serve as a good basis for evaluating the exchange of knowledge between the current interviewed community and its semi-nomadic ancestors.

#### 3.4.1. Relative Frequency of Citation (RFC)

This index shows the local importance of each species [70], and it is given by the following formula.

RFC = FC/N; where FC is the number of informants mentioning the use of the species and N is the total number of respondents participating in the survey.

#### 3.4.2. The Informant Consensus Factor (ICF)

The Informant Consensus Factor (ICF) [70] is used to measure homogeneity on the informant’s shared knowledge regarding the use of plants to treat the different health disorder categories. This factor follows this formula ICF = (nur − nt) /(nur − 1) where “nur” mentions the number of use reports for each use and “nt” refers to the number of taxa used to relieve the same ailment.

The value of this factor ranges from 0 to 1. A high value (close to 1) indicates that relatively few species are used by a large proportion of respondents, while a low value indicates that the informants disagree on the taxa to be used in the treatment within an ailment category.

In this study, diseases recorded in our survey were regrouped into 17 main categories reflecting local health problem classification.

#### 3.4.3. Ethnobotanicity Index (EI)

To estimate the importance of the used taxa in the studied territory, we used the Ethnobotanicity Index (EI) postulated by Porteres in [71] that stands for the ratio between the useful reported plant species and the total flora of the study area expressed as a percentage.

## 4. Conclusions

The floristic surveys carried out from 2014 to 2019 have enabled us to inventory 165 plant taxa of which three are endemic to Tunisia (*T. alopecurus*, *Teucrium sauvagei* Le Houér. and *A. campestris* subsp. *cinerea*), 32 are endemic to Maghreb, and/or North Africa, and 18 are widespread plants taxa that shape the main steppe vegetation of the area. Such results outline the richness and diverse flora of Ouled Dabbeb sector despite its small size (100 km^2^) and ecological harshness.

As regards endemic, rare, or threat wild taxa namely *L. coronopifolia*, *A. roseum* subsp. *Odoratissimum*, and *R. eriocalyx*, it is suggested to launch specific actions to verify if they are Crop Wild Relatives (CWR) and which is their genetic affinity with the respective parental species [72]. CWR taxa are potential sources of traits beneficial to crops, such as pest or disease resistance, yield improvement, or stability [73], and once identified, there is an imperative to effectively conserve these critical taxa in situ (i.e., in natural habitats managed as genetic reserves) and ex situ (primarily as the seeds in gene banks or as mature individuals in field collections) to underpin future world food [74,75].

Considering the ethnobotanical study, collected data indicated that the age and level of education were the main factors that seem to impact respondents’ ethnobotanical knowledge, which was also significantly patterned by the interaction between gender and use categories and well related to gender roles among inhabitants. This points out how traditional knowledge is embedded in social norms despite the challenging threat of being eroded and lost for the posterior.

This study revealed also that the local community of Ouled Dabbeb perceive that 70 plant taxa belonging to 59 genera from 31 families are mainly used for therapeutic (83%), food (49%), domestic (15%), ethnoveterinary (12%), cosmetic (5%), and ritual purposes (3%), bearing in mind that a single taxon could be useful for various purposes. Therefore, this ascertainment highlighted the wealth and wide-ranging of ethnobotanical knowledge that is still sustained by local people. Furthermore, 73% of reported ethnospecies were freely gathered from the wild, which points out the high inhabitant awareness of the ethnobotanical importance of the available surrounding flora. This result underpins local populations’ resilience and their adaptive capacity, which should be valued and protected as a part of biocultural diversity conservation strategies.

Interestingly, a high quantitative ethnobotanical index value has confirmed the versatility and widespread use of seven endemic plant taxa namely *A. campestris* subsp *cinerea*, *F. tunetana*, *A. saharae*, *R. suaveolens*; *S. tripartita*; *A. armatus*, and *H. deserti*. Such a finding strengthens the originality of the recorded folk knowledge. However, it cannot be ruled out that the proclivity of these endemics makes them more vulnerable to overexploitation, especially those whose non-renewable parts (underground storage organs and roots) are used.

Finally, this work brings out valuable and novel information about 11 new useful plants and 16 taxa with new utilization that have not been recorded previously in Tunisian ethnobotanical bibliography including mainly the use of *D. tortuosa* as biopesticide, *M. arvensis* as anticancer treatment, and the high use of *L. coronopifolia*, which is a new species that is reported for the first time in this area. Therefore, this study sets a starting point and basis for subsequent pharmacological, biological, and phytochemical investigations on taxa with high use values and then the discovery of many bioactive principles.

## Figures and Tables

**Figure 1 plants-10-00642-f001:**
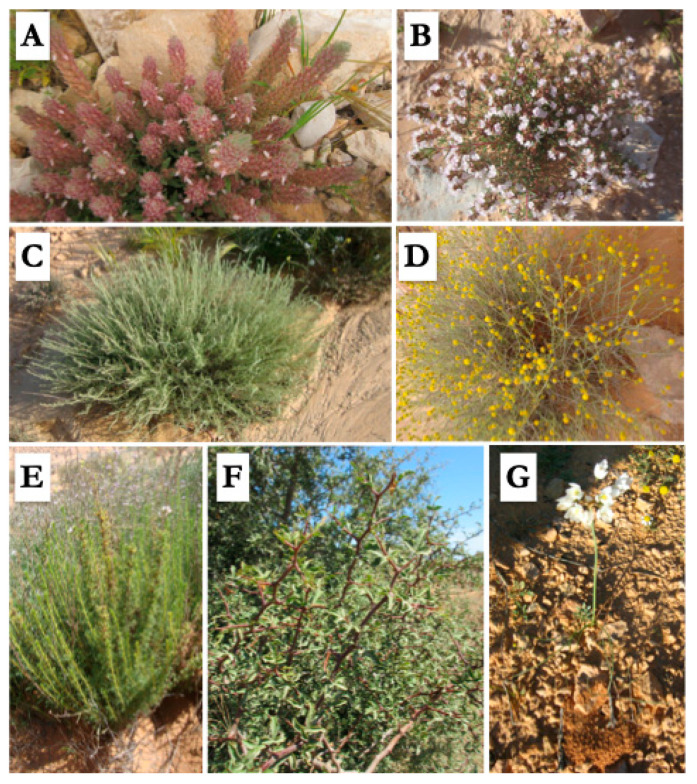
Some endemic taxa inventoried in the study area. Photos (**A**): *Teucrium alopecurus* de Noé; (**B**): *Thymus algeriensis* Boiss. & Reut.; (**C**): *Artemisia saharae* Pomel; (**D**): *Rhanterium suaveolens* Desf.; (**E**): *Artemisia campestris* L. subsp. *cinerea* Le Houér; (**F**): *Searsia tripartita* (Ucria) Moffett; (**G**): *Allium roseum* L. subsp. *odoratissimum* (Desf.) Murb.

**Figure 2 plants-10-00642-f002:**
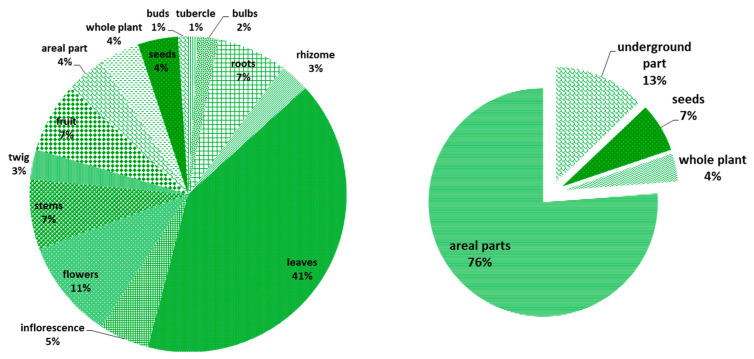
Plant parts used (%).

**Figure 3 plants-10-00642-f003:**
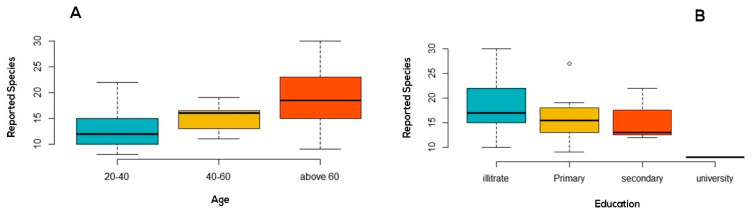
Informant’s competence according to the number of ethnospecies he/she mentioned by (**A**): Age and (**B**): Education Level.

**Figure 4 plants-10-00642-f004:**
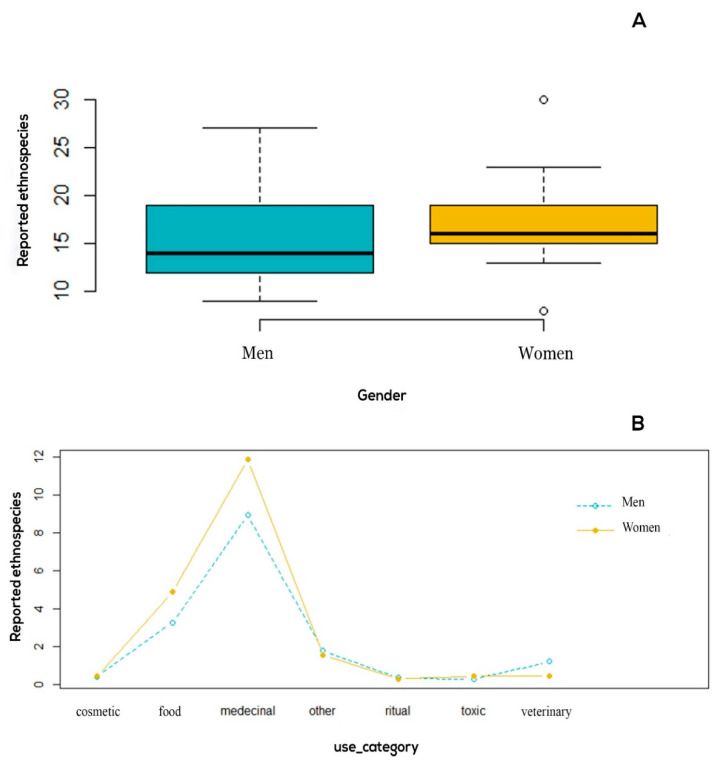
(**A**) Number of cited ethnospecies variations according to use category and gender obtained from the ANOVA analysis. (**B**) Informants competence according to the number of ethnospecies he/she mentioned by gender.

**Figure 5 plants-10-00642-f005:**
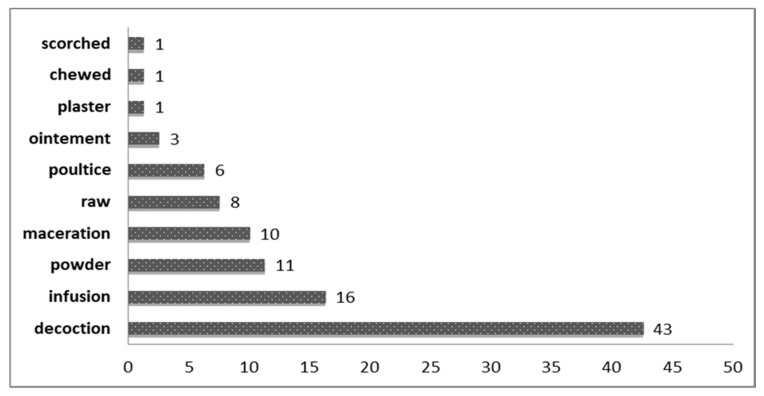
Use of reported ethnospecies according to method of preparation.

**Figure 6 plants-10-00642-f006:**
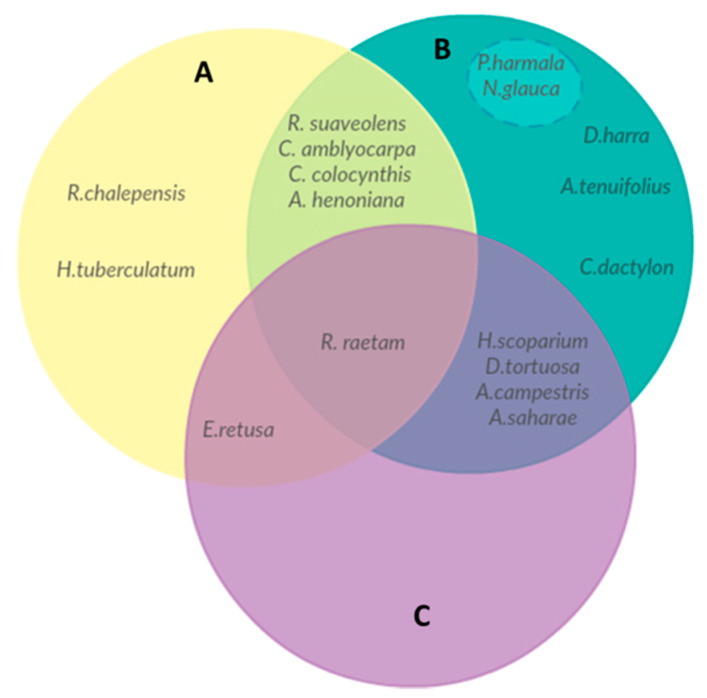
Overlap between (**A**) toxic taxa reported in this study, (**B**) taxa used by local communities of Ouled Dabbeb and reported by Le Floc’h [13] as toxic, and (**C**) plants reported as toxic but cautiously used.

**Figure 7 plants-10-00642-f007:**
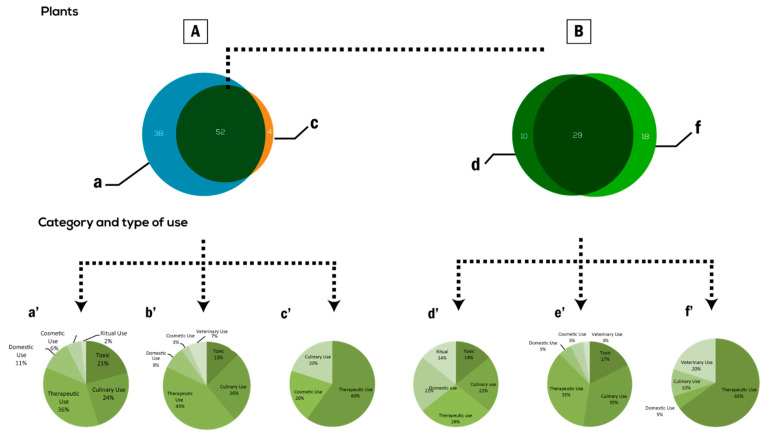
Comparison of ethnobotanical knowledge reported by the respondents in the present study and their forefathers reported by Le Floc’h [13]. (**A**) Venn’s diagram showing the shared ethnobotanical taxa between (**a**) the current community of Ouled Dabbeb and (**c**) their semi-nomadic ancestors reported by Le Floc’h [13]; (**B**) Venn’s diagram showing the shared plant taxa cited as useful by both the respondents in this study and their ancestors reported by the Floc’h [13] with d’: different method of preparation only cited by Le Floc’h [13] (**d**) and d: novel use only cited in this study (**e**); **a’, b’, c’, d’, e’** and **f’** represent the category and type of uses of taxa in **a, a ∩ c, c, d, d ∩ e**, and **e** respectively.

**Figure 8 plants-10-00642-f008:**
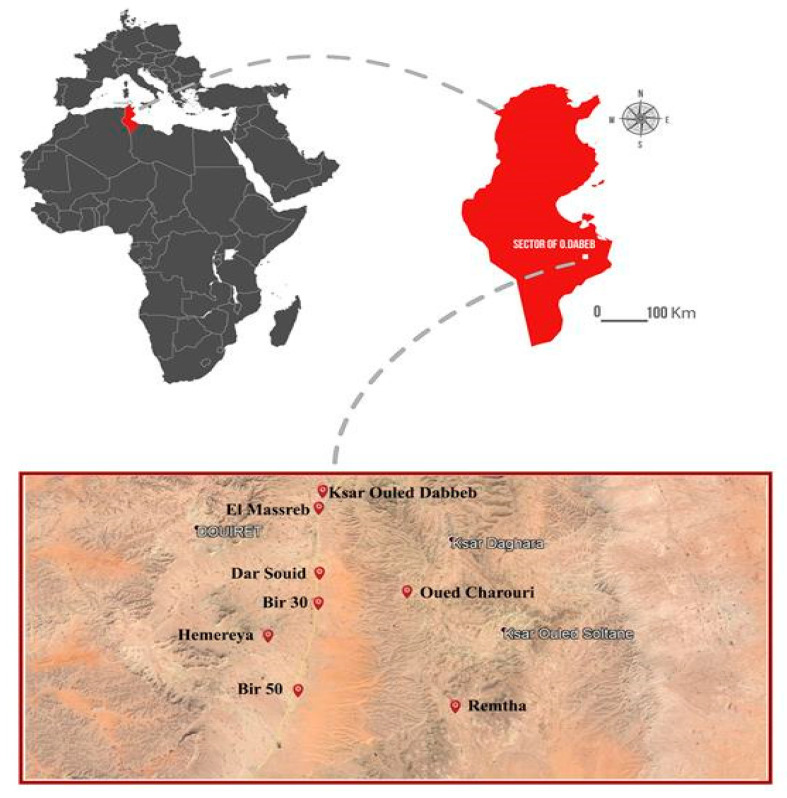
Geographical location of sector Ouled Dabbeb, the study (WGS 84/UTM zone 28N).

**Table 1 plants-10-00642-t001:** Taxa used in Ouled Dabbeb, Southern Tunisia with whole description of their use (family, taxon, vernacular name, hazard categoy, endemism, growth form, habitat, plant part used).

Taxon ‘Vernacular Name(s)’ (Herbarium Specimen)	Threat Categories	Endemism	Growth Form	Type of Vegetation	Plant Part(s) Used	Use Category	RFC
						Category and Type of Use, Preparation, and Administration	
*Allium ampeloprasum* L. ‘Korrath’ INAT3505	_	_	Herb	4	Leaves Bulbs	**Med:** Bulbs macerated in oil are orally taken as antibiotic and expectorant in cold and to treat coughs and sore throats. **Al:** Raw leaves and bulbs are incorporated in the traditional flatbread “Kesra”.	0.2
*Allium roseum* L. subsp. *odoratissimum* (Desf.) Murb. ‘Yazoul’, ‘Lazoul’ INAT3506	_	Endemic of North Africa	Herb	4	Leaves Bulbs	**Al**: Raw leaves are used to prepare a traditional flatbread “kesra” and can also be stewed and added to couscous or used as a condiment, replacing sometimes the onion.	0.5
*Searsia tripartita* (Ucria) Moffett ‘Jderi’ INAT3507	Rare	Endemic of Northern Sahara	Shrubby Tree	1	Roots	**Med:** Decoction of peeled roots is used as digestive and to treat ulcers **AL/Med:** To relieve ulcers (powder of peeled roots) added to a traditional dish (ich). **Dom**: Powdered roots barks are used for tanning leather and dyeing it to light brown.	0.4
*Ferula tunetana* Pomel ex Batt. * ‘Kalkh’ INAT3508	_	Endemic of Tunisia	Herb	-	Rhizome	**Med:** Decoction is orally taken to improve women’s fertility. **Al/Med:** Mineralizing and anti-inflammatory for bones and joints (rhizome powder added to “ich”).	0.3
*Deverra denudata* (Viv.) Pfiesterer & Podlech ‘Oljen’ INAT3509	According to the IUCN criteria this western and northern saharian species falls into the “C” category	_	Sub Shrub	2,3,4	Stems Flowers	**Dom:** It is used as an insect repellant, as odor control, and a deterrent against tadpoles in ‘Magel’, which is a sort of traditional Tunisian cisterns for collection and storage rainwater (hang a bunch of twigs).	0.1
*Deverra tortuosa* DC. ‘Gazzeh’ INAT3510	_	_	Shrub	3,4	Stems Leaves Flowers	**Med:** It is used to treat gas trouble (leaves infusions) and also used as antidiabetic (aerial part decoction) and to relieve hepatitis (infusion or water decoction of leaves and flowers). **Dom:** It is used as odor control, and a deterrent against tadpoles in ‘Magel’ (hanging a bunch of stems). It is also against insects and plant pests (incorporating stems’ powder into the soil).	0.5
*Periploca angustifolia* Labill. ‘Holleb’ INAT3511	_	_	Shrub	2	Leaves Bark of roots	**Med:** Water decoction of leaves is orally taken as hypotensive. **AL**: Root bark is appreciated when added to black tea.	0.2
*Asphodelus tenuifolius* Cav. ‘Tazia’, ‘Guitout’ INAT3512	_	_	Herb	2	Leaves	**Al:** Leaves are freshly cooked as part of the sauce for couscous and also to prepare traditional soup ‘dchich’.	0.3
*Artemisia arborescens* L. ‘Chajret Mariem’ INAT3513	_	_	Shrub	-	Leaves Flowers	**Med:** Water infusion is used as hypotensive.	0.0
*Artemisia campestris* L. subsp. *cinerea* Le Houér. ‘Tgoufet’ INAT3514	_	Endemic of Tunisia	Bush	1,2,3,4	Twigs Leaves	**Med:** Powdered leaves are mixed with milk to concoct a poultice that is applied on the leg fracture as plaster. A decoction of twigs is used as bechic. Maceration in vinegar is used as febricide and to alleviate leg pains. Leaves powder is sprinkled with olive oil to make a poultice that is applied against snake bites and scorpion stings. This poultice is also applied on the forehead to relieve cold. **Ethnovet:** It is used as a plaster for injured livestock (twig powder with milk).	0.8
*Artemisia saharae* Pomel ‘Chih’ INAT3515	_	Endemic of Magreb	Herb	1,2,3,4	Twigs Leaves	**Med:** Poultice made by leaves and lukewarm olive oil is tied to the forehead for headaches as a febricide. Leaves’ decoction is orally taken as hypotensive **, to relieve stomach pains and also as bechic. **Al:** Twigs are added to the tea as a flavoring. **Dom:** It used as repulsive of flies and other insects, as odor control, and a deterrent against tadpoles by hanging a bunch of twigs above ‘Magel’.	0.7
*Launaea quercifolia* Pamp. ‘Zarset Azouza’ INAT3516	_	Endemic of Sahara	Herb	4	Leaves	**Al:** Freshly cooked to prepare stew.	0.0
*Podospermum laciniatum* subsp. *decumbens* (Guss.) Gemeinholzer & Greuter ‘Telma’ INAT3518	_	_	Herb	2,4	Leaves Roots	**Med:** Leaves decoction is used to relieve stomach pains. **Al:** Leaves are freshly cooked as ingredients of couscous/ leaves are used raw in salads. **Dom:** Peeled roots used to make a ‘harkous’: a kind of temporary tattoo.	0.1
*Otoglyphis pubescens* (Desf.) Pomel ‘Wezweza’ INAT3517	According to the IUCN Criteria this north African endemic species falls into the E category	Endemic of North Africa	Herb	3	Whole plant	**Med:** Water decoction is orally taken to relieve kidney pains, to soothe varicella, and itching (compress with infusion water).	0.1
*Rhanterium suaveolens* Desf. ‘Arfej’ INAT3519	_	Endemic of Magreb	Shrub	2,3,4	Flowers Aerial parts	**Med:** It is used to relieve urinary incontinence (decoction of aerial parts). ** **Dom:** Water decoction of flowers is used to dye the wool in yellow. ** **Ethnovet:** It is used as diuretics for livestock (aerial parts’ decoction). **T**: It can be toxic to livestock if eaten in summer.	0.4
*Scorzonera undulata* Vahl ‘Guiz’ INAT3520	_	_	Herb	2, 4	Fleshy roots Flowers Leaves	**Al:** Flowers (having the taste of chocolate), leaves, stems, and fleshy roots are often eaten at the site of collection, as a snack.	0.3
*Diplotaxis harra* Boiss. ‘Harra’, ‘Choltam’ INAT3521	_	Endemic of North Africa	Herb	2,3	Leaves	**Al:** Leaves are freshly cooked to prepare stew.	0.1
*Eruca pinnatifida* Pomel ‘Harret El Bel’ INAT3522	_	_	Herb	2,3	Leaves	**Med:** Chewed leaves are used to relieve teeth pains.	0.0
*Moricandia arvensis* (L.) DC. ‘Hmim’ INAT3523	_	_	Herb	2,3	Leaves Flowers Roots	**Med:** Leaves decoction is used to relieve stomach pains, gas trouble, ulcers, and constipation. Leaves and flowers’ infusion is used as hypotensive and antidiabetic. Compress with roots water infusion is used to relieve dermatoses **Al/Med:** Leaves and flowers are freshly cooked as part of a traditional dish (a sort of thick soup) often prepared by local people of Tataouine in spring which is considered as a panacea, to increase immunity, to treat or prevent the development of cancer **.	0.7
*Henophyton deserti* Coss. & Durieu ‘Alga’ INAT3524	_	Endemic of Sahara	Bush	2,3	Leaves	**Med:** Water infusion of leaves is used to improve women’s fertility **.	0.1
*Capparis spinosa* L. ‘Kabbar’ INAT3525	_	_	Shrubby Tree	2	Leaves Flowers Buds Fruits	**Med**: Poultice made from crushed leaves is applied on the forehead to relieve fever. A decoction of leaves and/or flower buds is used as antidiabetic and diuritic. **Al:** Leaves and fruits are freshly cooked to prepare main dishes in the region couscous, stew, and soup.	0.6
*Cleome amblyocarpa* Barratte & Murb. ‘Mnitna’ INAT3526	_	_	Herb	3	Leaves	**Med:** A decoction is used as a bechic and sedative. Water infusion is used to improve women’s fertility **. **T**: It is a toxic plant that causes nervous disorders in animals.	0.0
*Atriplex glauca* L. *Atriplex halimus* L. *Atriplex mollis* Desf. ‘Gtaf’ INAT3527, INAT3528, INAT3529	_	*Atriplex halimus* L. Endemic of Maghreb	Shrubby tree/Shrub/Bush	2,3	Leaves	**Med:** Fresh leaves macerated in oil are applied on the affected area (as a bandage) as antinflammatory and antirheumatic and against joint pains **, leaves are eaten raw to relieve constipation. **Al:** Leaves are used raw in salads/ freshly cooked as part of couscous/dried leaves are ground and used as a replacement or complement to barley flour to make bread (in times of food insecurity in the past).	0.0
*Haloxylon scoparium* Pomel ‘Remth’ INAT3530	_	_	Shrubby tree	2,3	Twigs Leaves	**Med:** The poultice is applied locally on the skin for wounds healing and as hemostatic. It is also used as a disinfectant (washing with water leaves infusion) and as collyrium (fresh twigs’ water infusion). **Ethnovet:** Water infusion is applied locally on the skin of livestock against scabies and as disinfectants after mowing, infusion (drink) as appetizer, and digestive. **Dom**: It is used to prepare snuff powder ‘neffa’ to which it owes its degradation.	1.0
*Salsola acanthoclada* Botsch. * ‘Souida’ INAT3531	_	_	Subshrubs/Shrubs,	2	Stems	**Med:** Water decoction is used to alleviate lumbar pains.	0.0
*Helianthemum confertum* Dunal *Helianthemum lippii* (L.) Dum.Cours. ‘Chaal’ INAT3532, INAT3533, INAT3534	_	_	Shrub	2	Aerial parts	**Med:** Poultice (powdred dried aerial parts with oil) is applied locally on the skin to treat wounds as hemostatic. **O:** Nearby, these taxa truffles grow up.	0.2
*Citrullus colocynthis* (L.) Schrad. ‘Handhel’ INAT3535	_	_	Herb	2,3	Peeled fruits Sarcocarps Seeds	**Med:** It helps to alleviate rheumatic pains (scorched peeled fruit is crushed with pain leg), to relieve hemorrhoid (wraps with macerated sarcocarps in oil or friction with scorched sarcocarps). ** Seeds decoction is used to treat scorpion bites and dermatoses (compress). **T**: It can cause death if it is orally taken.	0.5
*Juniperus phoenicea* L. ‘Arar’ INAT3536	_	_	Shrub	1,3	Leaves	**Med:** Decoction/infusion with *Ruta chalepensis* L., *Thymus algeriensis* Boiss. & Reut., and *Rosmarinus eriocalyx* Jord. & Fourr. is used to relieve coughs. Powder of leaves are eaten with egg to treat diarrhea.	0.4
*Cynomorium coccineum* L. ‘Tarthouth’ INAT3537	_	_	Herb	3	Roots Aerial parts	**Med:** To relieve stomach pains and ulcers (powder of dried plants eaten with a cup of water). ** and to treat common colds (water decoction of roots).	0.2
*Euphorbia retusa* Forssk. ‘Lebbina’ INAT3538	_	_	Herb	3	Roots	**Med:** Maceration in oil is applied on the skin against eczema and spot. ** **T:** A direct application on skin can be toxic and induce an acute inflammation.	0.0
*Ricinus communis* L. ‘Kharwae’ INAT3539	_	_	Shrub	-	Leaves	**Med:** It used for cold and mild respiratory (soaked fresh leave in olive oil and put on the thorax).	0.0
*Anthyllis henoniana* Batt. ‘Ghazdir’ INAT3540	_	_	Bush	2,4	Flowers	**T**: Toxic for livestock after eating flowers.	0.1
*Retama raetam* (Forssk.) Webb & Berthel. ‘R’tem’ INAT3541	_	_	Shrubby tree	2,4	Leaves	**Med:** Leaves powder is sprinkled with olive oil to make a poultice that is locally applied on the skin to treat wounds as hemostatic to relieve dermatoses, furunculosis, eczema, spot, scorpion bites, and also to relieve headache when powdered leaves are added to henna. **Ethnovet:** Fresh leaves soaked in milk are tied to injured leg as plaster. **T:** It can cause death if it is orally taken.	0.7
*Trigonella foenum-graecum* L. ‘Helba’ INAT3542	_	_	Herb	-	Seeds Aerial parts	**Med:** Seeds are orally taken with a cup of water to relieve diarrhea. **Al**: Leaves freshly cooked as part of couscous.	0.0
*Astragalus armatus* Willd. * ‘Gtet’ INAT3543	_	Endemic of Maghreb	Shrub		Roots	**Med:** A decoction of roots is used to relieve odentonecrosis (Mouthwash).	0.1
*Calicotome villosa* (Poir.) Link * Gueddim INAT3544	_	_	Shrubby tree		Leaves	**Med:** Water decoction of fresh leaves is orally taken to alleviate kidney pains. **Dom:** Leaves are used to concoct basketry.	0.1
*Erodium glaucophyllum* (L.) L’Hér. Tommir/Merghid INAT3545	_	_	Herb		Tubercles	**Med:** Tubercles are eaten raw to relieve stomachache **** Al:** Tubercles are eaten raw. Fresh leaves are used raw in salads/cooked as part of couscous.	0.3
*Globularia alypum* L. Zriga INAT3546	_	_	Sub -Shrub		Leaves	**Med:** Crushed leaves with milk (ointment) is used to treat furunculosis. Decoction is orally taken to relieve urinary incontinence.	0.1
*Teucrium alopecurus* de Noé * ‘Jaâda’ INAT3547	_	Endemic of Tunisia	Sub -Shrub		Leaves Flowers	**Med:** Decoction is orally taken as antidiabetic and hypotensive. It is used for postpartum hygiene and to relieve stomach ache; powder of flowers is locally applied (compress). Maceration in oil is applied on the skin against eczema and spot.	0.2
*Rosmarinus eriocalyx* Jord. & Fourr. * ‘Klil’ INAT3548	_	_	Shrubby tree	1	Leaves	**Med:** It helps to relieve cough, fever, and cold (decoction/infusion), smoke to treat rheumatism. **Al**: Condiment.	0.6
*Ajuga iva* (L.) Schreb. ‘Chendgoura’ INAT3549	_	_	Herb	1	Leaves	**Cosm:** Maceration in oil is applied on hair as softer. **Med:** Decoction is used as emmenagogue, as hypotensive, and antidiabetic.	0.4
*Marrubium alysson* L. ‘Murrûbya’ INAT3550	_	According to the IUC criteria this endemic species falls into the “ EN” category	Herb	3	Leaves Stems	**Med:** Decoction (drink) as bechic.	0.1
*Salvia aegyptiaca* L. ‘Kammouna’ INAT3551	_	_	Herb	1,3	Seeds Flowers	**Al:** Seeds are used as flavoring in bread. **Dom:** Flowers are used to perfume hair (maceration in oil).	0.1
*Lavandula coronopifolia* Poir. *Lavandula multifida* L. ‘Khzema’ ‘Kammoun’ INAT3552, INAT3553	_	_	Shrub	3	Leaves Flowers	**Med:** Decoction of leaves is used as emmenagogue, as hypotensive, and antidiabetic. **Cosm**: Maceration of flowers in oil is applied on hair as softer and flowers are used to prepare a hair perfume.	0.4
*Mentha pulegium* L. ‘Flayou’ INAT3554	_	_	Herb	3	Leaves	**Med:** Rinses and wraps with infusion water/water extract are used to treat common colds, fever, and also to relieve headache.	0.1
*Malva aegyptia* L. ‘Khobbiza’ INAT3555	_	_	Herb	3	Leaves	**Med:** Water decoction is orally taken to relieve gas pains and stomachache. Water infusion (compress) is applied topically to relieve hemorrhoids ** **Al:** Freshly cooked to prepare stew.	0.2
*Myrtus communis* L. ‘Jedra’ INAT3556	_	_	Shrub	-	Leaves Buds	**Med:** Water infusion of leaves is orally taken to relieve headaches ** or poultice of crushed leaves with heated oil. **Cosm**: Macerated in oil buds are used to prepare a homemade perfume	0.1
*Nitraria retusa* Asch. ‘Ghardag’ INAT3557	_	_	Shrub		Leaves	**Med:** Decoction in milk of powdered dried leaves is used to prepare ointment that is applied to treat leishmaniasis and furunculosis **.	0.1
*Olea europaea* L. ‘Zitoun’ INAT3558	_	_	Tree	2,3	Twigs Leaves Fruits	**Med:** A decoction of leaves andtwigs is orally taken as hypotensive, antidiabetic, as antiseptic, and to relieve teeth pains. **Dom:** Black olives were used for dyeing wool in pinkish white.	0.4
*Peganum harmala* L. ‘Harmel’ INAT3559	_	_	Herb		Seeds Aerial parts	**Med:** It is used as a panacea. Seeds are directly eaten with a cup of water as digestive, to relieve stomach pains and also as sedative. It is used to relieve hemorrhoids (aerial parts infusion orally taken), to alleviate feet pains given by swelling or fatigue (footbath with leaves’ infusion), as eye inflammation remedy (compress), and also as febricide (wraps).	0.4
*Plantago albicans* L. ‘Inam’ INAT3560	_	Endemic of Sah of north Africa	Herb	1,2,4	Leaves	**Med:** Water infusion is used to relieve urinary incontinence. **Ethnovet:** Water infusion of leaves is also diuretic for livestock, and it is also used to recover the appetite or to avoid indigestion.	0.3
*Stipagrostis pungens* (Desf.) de Winter * ‘Sbat’ INAT3561	_	_	Herb	4	Inflorescences	**Med:** Water decoction of inflorescence is used to relieve urinary incontinence.	0.1
*Cynodon dactylon* (L.) Pers. ‘Najm’ INAT3562	_	_	Herb	4	Aerial parts	**Med:** It used to relieve urinary incontinence (water decoction) and as a sedative (water decoction with *Polygonum equisetiforme* Sm.	0.1
*Lycium shawii* Roem. & Schult. * ‘Sakkoum’ INAT3563	_	_	Shrub	4	Leaves Fruits	**Med:** It is used to relieve constipation (raw fruit and water decoction of leaves).	0.1
*Polygonum equisetiforme* Sm. ‘Gorthab’ INAT3564	_	_	Herb	3	Leaves	**Med:** Mushed fresh leaves or powdered dried ones are used to prepare poultice that is applied locally on the skin to treat wounds as hemostatic, and also as a disinfectant (washing with water infusion. **Ethnovet**: Leaves’ powder is mixed with milk to concoct a poultice that used as plaster for livestock.	0.3
*Rumex roseus* L. ‘Hommitha’ INAT3565	_	_	Herb	2,3	Leaves Flowers	**Al:** Leaves and flowers are used raw in salads.	0.0
*Calligonum azel* Maire ‘Azel’ INAT3566	_	Endemic of Magreb	Shrub	4	Leaves	**Ethnovet:** Wraps with leaves’ infusion are used for the treatment of scabies of camels.	0.0
*Calligonum polygonoides* L. ‘Arta’ INAT3567	_	_	Shrubby tree	4	Roots Leaves	**Med:** Water decoction of roots is used as anthelmintic. **Ethnovet:** It is used for the treatment of scabies of the camel (wraps with leaves infusion).	0.0
*Punica granatum* L. ‘Rommen’ INAT3568	_	_	Tree	-	Bark of fruits Flowers	**AL/Med:** Powder of dried fruit’s bark is used to relieve ulcers when added to traditional dish (ich) ** **Dom:** Dyeing wool in orange with flowers and fruit set locally called “Lallouch”.	0.1
*Ziziphus lotus* Lam. ‘Sedr’ INAT3569	_	_	Tree	1,2, 4	Fruits	**Al:** Fruits are eaten (fresh/dried) as a snack. **R:** Thanks to their saponifynig properties, the malikites use these fruits to wash and shroud the deceased.	0.3
*Ruta chalepensis* L. ‘Fijel’ INAT3570	_	_	Herb	2	Leaves Twigs	**Med:** To relieve otitis (drops of macerated leaves in oil). It is also orally taken for postpartum hygiene as a hypotensive and as a painkiller **R:** In fumigation, the twigs are used to treat epilepsy and bad eyes. **T:** Abortive plant.	0.5
*Haplophyllum tuberculatum* (Forssk.) A.Juss. ‘Offina’ INAT3571	_	_	Perennial herb	2,3	Aerial parts	**Med:** Decoction is used as hypotensive. **T:** Abortive plant.	0.1
*Nicotiana glauca* Graham ‘Hchichet Erih’ INAT3572	_	Endemic of North Africa	Shrub	3	Leaves	**Med:** Water decoction is used to relieve gas pains **.	0.1
*Thymus algeriensis* Boiss. & Reut. ‘Zaater’ INAT3573	_	Endemic of Magreb	Herb	1,3	Leaves	**Med:** Water infusion helps to relieve coughs and cold. Decoction is orally taken as febricide **Al:** Condiments.	0.4
*Thymelaea hirsuta* Endl. * ‘Methnen’ INAT3574	_	_	Shrubby Tree	3	Whole plant	**Med**: To relieve sciatica (dried plant is trampled by a painful leg) **.	0.1
For Type of vegetation: 1: Limestone Mountain Steppe with *Macrochloa* *tenacissima* Kunth, *Thymus algeriensis* Boiss. & Reut., and *Genista microcephala* Coss. & Durieu 2: Steppe with *Artemisia herba-alba* Asso and *Haloxylon scoparium* Pomel that characterised loamy arid piles of the study area. 3: Steppe with *Thymus capitatus* Hoffmanns. & Link and *Artemisia campestris* L. that defines the wadi beds. 4: Steppe of sandy plains with *Rhanterium suaveolens* Desf., *Anthyllis henoniana* Batt., and *Astragalus armatus* Willd.							

**RFC**: Relative frequency of citation, **MED**: Medicinal use, **AL**: Alimentary use, **Ethnovet**: Veterinary use, **Dom**: Domestic use; **Cosm**: Cosmetic use, **R**: Ritual use, **T**: Toxic, **‘-’**: No data are available, *****: New reported ethnobotanical taxa, ******: New reported uses.

**Table 2 plants-10-00642-t002:** Sociodemographic characteristics of informants and number of cited ethnospecies by gender, age, education level, and location.

Sociodemographic Variables		Number of Informants (%)	Number of Cited Ethnospecies (%)	*p* Value
**Age (years)**	20–40	5 (16.66)	23 (34.85)	0.048 *
	40–60	11 (36.66)	44 (66.67)	
	>60	14 (46.66)	59 (89.39)	
**Gender**	Male	14 (46.66)	55 (83.33)	0.060
	Female	16 (53.33)	39 (59.09)	
**Location**	Rural	24 (80)	58 (87.88)	0.133
	Urbain	6 (20)	42 (63.64)	
**Education level**	Illiterate	14 (46.66)	47 (71.21)	0.046 *
	Primary	11 (36.66)	34 (51.52)	
	Secondary	4 (13.33)	22 (33.33)	
	University	1 (3.33)	8 (12.12)	

Values with * indicate the significant correlation of the number of cited ethnospecies with the sociodemographic variable (* *p* < 0.05).

**Table 3 plants-10-00642-t003:** Number of use report (Nur), ethnospecies (Nt), and values of the Informant Consensus Factor (ICF) per ailment category ordered from higher to lower ICF values.

Rank	Ailment Category	Nur	Nt	ICF
1	Cancer	8	1	1.00
2	Poisonous animal bites	12	3	0.82
3	Cold and respiratory tract diseases	42	9	0.80
4	Dermatological diseases	49	11	0.79
4	Bones diseases	15	4	0.79
5	Fever	33	9	0.75
6	Sexual reproductive problems	18	6	0.71
7	Gastrointestinal disorders	72	22	0.70
8	Cardiovascular diseases	29	10	0.68
9	Nervous system disorders	4	2	0.67
9	Pathologies ORL	4	2	0.67
10	Headache	15	6	0.64
11	Endocrine system diseases	14	6	0.62
12	Skeletomuscular diseases	10	5	0.56
13	Kidney and urinal tract diseases	20	10	0.53
14	Toothache and mouth inflammation	6	4	0.40
NR	Ophthalmological diseases	1	1	-

## Data Availability

Data is available upon appropriate request from the corresponding author.

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
