# Peer review of "Ethnobotanical Study on Plant Used by Semi-Nomad Descendants’ Community in Ouled Dabbeb—Southern Tunisia"

_plants, 2021, doi:10.3390/plants10040642_

Round 1

Reviewer 1 Report

Dear Authors,

I report here and in the attached files several important corrections to make to your manuscript and the Appendix A and B.

Please, correct graphical abstract with "Plant species". Check my notes.

Appendix A.

1) All species names must be followed by their authorship reported in normal, not italicized, and not followed by a comma unless necessary. Furthermore, even "subsp." it must be written not in italics. Please check the entire list. Check all authorships (ex.: "Desf.", not "Desf", and so on...).

2) Choose whether the various "Growth forms" should be written with a capital letter or not.

3) All the headings (Hazard categories, Synonyms, Growth form, Plant Life Form, Flowering period, Tunisia,  North africa, Maghreb Sahara) must be referenced in the Appendix and reported in Material and Methods section.

In the text

All scientific names must be reported complete with their authorship at the first mention in the text and after with genus name abbreviated.  Please, change "subsp" with "subsp.". Ex.: "Atriplex halimus L." at first mention and after A. halimus. Tables and figures are an exception as they must be self-explanatory, so you must always report the full name complete with its authorship. Ex.: Atriplex halimus L.

Sorry, but there are so many errors to correct that I can not follow with my review because in this form the manuscript is very flaw in its structure. For example, Table 1 is cutted and unreadable!

However, the work is very interesting and I encourage you to rewrite it and resubmit to the journal.

Best wishes.

Author Response

Responses to the reviewer’s comments

Manuscript Number: plants-1120357  

Title: Ethnobotanical study on plant used by semi nomad descendants’ community in Ouled Dabbeb –Southern Tunisia.

The authors would like to thank the reviewers for their thoughtful comments and valuable suggestions. Please find a point by point response to the reviewer’s comments:

Reviewer 1 comments:

Point 1: The authors took into consideration all the changes suggested by the reviewer. In reference to the recommendations, the following points were corrected:

Graphical abstract:

Point 2:

  • Thank you for pointing out this mistake, it has been corrected.

Abstract and keywords:

Point 3:

  • We added other keywords as suggested: traditional knowledge; wild food; medicinal plants; Ethnoveterinary; endemic; toxic; knowledge transmission. Please refer Page 2 line 30

Appendix A:

Point 4:

All species names must be followed by their authorship reported in normal, not italicized, and not followed by a comma unless necessary. Furthermore, even "subsp." it must be written not in italics. Please check the entire list. Check all authorships (ex.: "Desf.", not "Desf", and so on...). done

  • The requested recommendations were added in the Appendixes A and B tables as well as throughout the manuscript.

Point 4:

Choose whether the various "Growth forms" should be written with a capital letter or not.

  • Done:"Growth forms" are written with a capital letter

Point 5: All the headings (Hazard categories, Synonyms, Growth form, Plant Life Form, Flowering period, Tunisia,  North africa, Maghreb Sahara) must be referenced in the Appendix and reported in Material and Methods section.

  • The requested recommendations were added in the Appendixs A for as well as in Material and Methods section. Please refer Page 40 lines : 324 and 338.

In the text

Point 6:

  • Page 3 line 44 we replaced stage by belt as recommended

Point 7: Define the term of ethnospecies

  • Page 5 line 84 we have explained the term of ethnospecies as recommended

 Point 8: there is another heading with the same name 3.43. please change one of these 2

  • Page 9 line 131 we changed the title: Relevance of wild used plants in the sector of Ouled Dabbeb

Point 9: All scientific names must be reported complete with their authorship at the first mention in the text and after with genus name abbreviated.  Please, change "subsp" with "subsp.". Ex.: "Atriplex halimus L." at first mention and after A. halimus. Tables and figures are an exception as they must be self-explanatory, so you must always report the full name complete with its authorship. Ex.: Atriplex halimus L.

Sorry, but there are so many errors to correct that I can not follow with my review because in this form the manuscript is very flaw in its structure. For example, Table 1 is cutted and unreadable!

  • All scientific names are corrected as recommended in the Appendixes A and B tables as well as throughout the manuscript

Point 10: Page 37 line 270 we corrected the mistake as recommended: km 2

Point 12: Is this acronym in accordance with theirs 2020 if yes report it in Reference

  • Yes .Please refer to Page 40 Line 338 we added the reference of Herbarium code as described below

The fully herbarium specimens were coded and deposited in the herbarium of the National Agronomic Institute of Tunisia (INAT) (http://sweetgum.nybg.org/science/ih/herbarium-list/?NamOrganisationAcronym=INAT)

Reviewer 2 Report

The authors propose a manuscript titled “Ethnobotanical study on plant used by semi nomad descendants’ community in Ouled Dabbeb –Southern Tunisia”.

The work is extremely interesting and rich in original contents related on ethnobotanical species. However the presentation is not suitable for an international audience, it seems a part of a master thesis quickly summarized without giving the right weight to the data. I decided, in relation to the original data presented, to help the 2 authors to better present the work with the following suggestions:

Please pay attention and format title and whole document according to main document of Journal, and check english language.

Abstract and keywords: some period highlighted in green, why?

Abraviations. Please move in material and methods

Graphic abstract its not able to international audiance. Please cut

Introduction.

…are endemic in Tunisia not nationally endemic

Please move some specific comments in material and methods (to do), as… botanical surveys (2014-2019) …ethnobotanical surveys (2014, 2015 and 2016)…

Results and discussion

I suggested in order to have a better work to make a pictures of endemic species (if you personally have them)

Please % always next to the number with no spaces

high use of some therophytes

a short annual period that…

Tha authors declare. “We quoted the ethnobotanical use of 12 endemic and near endemic taxa”. There is no space for near endemic. The species is endemic or not endemic. Please correct in the suggested way.

When the authors cite for the first time the scientific name of the plant please add the name of the author that discover the species. An example: Artemisia campestris Turcz. subsp. cinerea Le Houér. Also check the correct scientific name: Artemisa campestris subsp. cinerea not cinera.

Table 1

Table.1: check the correct form. Please see the main document of the journal. The table is out of the merge of page. Please format all document, sometimes And and not and, cosm and not Cosm… nd other formal mistakes

In the first column cut the name of the family its not important in international audiance

Herbarium specimen not Herbarium voucher. Also please give not in the table but in material and methods the complete name of official herbarium specimen INAT52, INAT53…

The name of the  author not in italic. I.e. Allium ampeloprasum L. Please check all scientific name. Also subsp. not subsp

threat categories not Hazard categories

Habitat. The term wild is too generic as another words that the authors have used. Please specify the type of vegetation or at least if deciduous oak woods, Aleppo pine woods, evregreen sclerophyllous scrub, grassland, annual meadow….

Table 2

Please formatting

Figg 2 and 3 are not clear. Please change the risolution

The leaves and bulbs of the two Alliaceae, Allium. Please check the correct scientific name. For example: Allium roseum L. subsp. odoratissimum (Desf.) Murb. and not Allium roseum subsp. odoratissimum (Lazoul)

Move Material an methods before Results and discussion

Material and methods

The geographic position in WGS84? or other Geographic system? Please specify

After figure 7

Please complete the period in this way. “…agriculture devoted to sporadic cereal crop production and rustic arboriculture (olive and fig) and on livestock breeding (sheep, goats and camels). For the breeding of livestock it is desirable to provide a grazing plan in order to preserve natural habitats [Perrino et al. 2020], as better described below”.

Add a reference

Perrino, E. V., Musarella, C. M. & Magazzini, P. (2020). Management of grazing Italian river buffalo to preserve habitats defined by Directive 92/43/EEC in a protected wetland area on the Mediterranean coast: Palude Frattarolo, Apulia, Italy. Euro-Mediterr. J. Environ. Integr.(accepted) doi: 10.1007/s41207-020-00235-2.

The authors sentence: (1) Association with Stipa tenacissima and Artemisia herba alba dominated the surrounding mountains (2) association with Artemisia herba alba and Haloxylon scoparium char- acterised loamy arid piles of the study area. (3) Association with Rhanterium suaveolens and Artemisia campestris defined the sandy plains. (4) Association with Thymus capitatus and Artemisia campestris marked the wadi beds.

Please change the term association in grouping, because association is a specifc syntaxon not taxon. For example Pistacia pinetum halepensis. Please change in the suggested way: i.e. Stipa tenacissima and Artemisia herba grouping or specify the reative syntaxon.

The fully herbarium specimens not The fully Voucher specimens

Conclusion

Please complete with this crucial concept

After period “…sector despite its small size (100 km2) and ecological harshness”. Continue in this way:

As regard endemic or threat wild species, it is suggested to launch specific actions in order to verify if they are Crop Wild Relatives, and which is their genetic affinity with the respective parental species [Perrino and Wagensommer 2021, ].

Add these 3 references

Maxted, N.; Kell, S., Establishment of a Network for the In Situ Conservation of Crop Wild Relatives: Status and Needs. Commission on Genetic Resources for Food and Agriculture. Food and Agriculture Organization of the United Nations, Rome, Italy, 2009.

Zair, W.; Maxted, N.; Brehm, J. M.; Amri, A., Ex situ and in situ conservation gap analysis of crop wild relative diversity in the Fertile Crescent of the Middle East. Genet. Resour. Crop Evol., 2020. doi: 10.1007/s10722-020-01017-z

Perrino, E. V., Wagensommer, R. P., Crop Wild Relatives (CWR) Priority in Italy: Distribution, Ecology, In Situ and Ex Situ Conservation and Expected Actions. Sustainability, 2021, 13, 4, 1682. https://doi.org/10.3390/su13041682.

Author Response

Responses to the reviewer’s comments

Manuscript Number: plants-1120357  

Title: Ethnobotanical study on plant used by semi nomad descendants’ community in Ouled Dabbeb –Southern Tunisia.

The authors would like to thank the reviewers for their thoughtful comments and valuable suggestions. Please find a point by point response to the reviewer’s comments:

Reviewer 2 comments:

Point 1:Please pay attention and format title and whole document according to main document of Journal, and check english language.

  • The grammatical errors were carefully corrected throughout the manuscript. The English language of the manuscript has been improved.

Point 2: Abstract and keywords: some period highlighted in green, why?

  • Thank you for pointing out this mistake, it has been corrected.

Point 3: Abraviations. Please move in material and methods

  • We removed abreviations from the text because according to 'Instructions for Authors' on the journal website the abbreviations have been defined in parentheses the first time

they appear in the abstract, main text which is revised throughout the manuscript

Point4: Graphic abstract it’ s not able to international audiance. Please cutThe sentence was completed as suggested.

  • Thank you for pointing out this it has been corrected according to  'Instructions for Authors' on the journal website. We have made some modifications and we corrected some English mistakes as Plant species.

Point 5: Introduction.

…are endemic in Tunisia not nationally endemic

  • Thank you for pointing out this. The correction was made. Please refer to Page3 Line 36 we have changed nationally endemic by endemic to Tunisia

Please move some specific comments in material and methods (to do), as… botanical surveys (2014-2019) …ethnobotanical surveys (2014, 2015 and 2016) done

  • Thank you for pointing out this. The correction was made. Please refer to Page4 Lines: 57 and 59.

Point 6: Result and discussion.

I suggested in order to have a better work to make a pictures of endemic species (if you personally have them) I suggested in order to have a better work to make a pictures of endemic species (if you personally have them) The name of the professor was removed as suggested.

  • .Thank you for suggestion the we made a picture of 7 endemic species Page 8

Point 7: Please % always next to the number with no spaces.

  • Thank you for pointing out this. This mistake was corrected throughout the manuscript.

Point 8: high use of some therophytes done

a short annual period that…done

  • Thank you for pointing out this. The correction was made. Please refer to Page 6 Line 100-101

Point 9: Tha authors declare. “We quoted the ethnobotanical use of 12 endemic and near endemic taxa”. There is no space for near endemic. The species is endemic or not endemic. Please correct in the suggested way.

  • Thank you for pointing out this. The correction was made. Please refer to Page7 Line 109

Point 10: When the authors cite for the first time the scientific name of the plant please add the name of the author that discover the species. An example: Artemisia campestris Turcz. subsp. cinerea Le Houér. Also check the correct scientific name: Artemisa campestris subsp. cinerea not cinera.

  • All the scientific name of the plant were corrected as recommended in the Appendixes A and B tables as well as throughout the manuscript.

Table 1

Point 11: Table.1: check the correct form. Please see the main document of the journal. The table is out of the merge of page. Please format all document, sometimes And and not and, cosm and not Cosm… nd other formal mistakes

  • Thank you for pointing out this. The correction was made. Please refer to Page 11 to 20

Point 12: In the first column cut the name of the family it’s not important in international audience

  • Thank you for pointing out this. We removed the name of family from the first column of table . Please page 11,

Point 12: Herbarium specimen not Herbarium voucher. Also please give not in the table but in material and methods the complete name of official herbarium specimen INAT52, INAT53…

  • Thank you for pointing out this. The correction was made in the table 1 as well in the texte. Please refer to Page 11 and 40 Line 338 as below :

The fully herbarium specimens were coded and deposited in the herbarium of the National Agronomic Institute of Tunisia (INAT) (http://sweetgum.nybg.org/science/ih/herbarium-list/?NamOrganisationAcronym=INAT)The fully herbarium specimens were coded and deposited in the herbarium of the National Agronomic Institute of Tunisia (INAT) (http://sweetgum.nybg.org/science/ih/herbarium-list/?NamOrganisationAcronym=INAT)

Point 13:The name of the  author not in italic. I.e. Allium ampeloprasum L. Please check all scientific name. Also subsp. not subsp

  • Thank you for pointing out this. The correction was made throughout the manuscript for all author as well for subsp .

Point 14: threat categories not Hazard categories

  • Thank you for pointing out this. The correction was made in The Table as well in the Appendix A. Please refer to Page 11

Point 15: Habitat. The term wild is too generic as another words that the authors have used. Please specify the type of vegetation or at least if deciduous oak woods, Aleppo pine woods, evregreen sclerophyllous scrub, grassland, annual meadow….

  • For this comment we have chosen to specify the type of vegetation for each plant species The correction was made in The Table as well in the Appendix A. Please refer to Page 11 and 20 in fact    we have 4 type of vegetation

1: Limestone Mountain Steppe with Macrochloa tenacissima (L.) Kunth., T. algeriensis and Genista microcephala Coss. Coss. & Durieu. 

2: Steppe with Artemisia herba alba Asso. and H. scoparium that characterised loamy arid piles of the study area.

3: Steppe with Thymus capitatus (L.) Hoffmanns. & Link. and A. campestris that defines the wadi beds.

4: Steppe of sandy plains with R. suaveolens A.henoniana and A. armatus

  •  

Point 16: Table 2

Please formatting

  • Done Please refer to Page 22

Point 17: Figg 2 and 3 are not clear. Please change the resolution

  • . Done Please refer to Page 23 and 24

Point 18: Move Material and methods before Results and discussion

  • According to 'Instructions for Authors' on the journal website Material and methods should be after Results and discussion

Point 19: Material and methods

The geographic position in WGS84?            or other Geographic system? Please specify

After figure 7.

  • Thank you for pointing out this. The correction was made. Please refer page 38 In fact we used WGS84/UTM zone 28 N.

Point 20: Please complete the period in this way. “…agriculture devoted to sporadic cereal crop production and rustic arboriculture (olive and fig) and on livestock breeding (sheep, goats and camels). For the breeding of livestock it is desirable to provide a grazing plan in order to preserve natural habitats [Perrino et al. 2020], as better described below”.

  • Thank you for this recommendation: from Line 289 to 293 we developed the suggested idea. hopefully we have well understood your comment:

camels). Though these agropastoral activities reflect a relative adaptation to the environmental conditions, it does not hide the great precariousness of this region, notably due to inappropriate use of pasture which could pose a threat for its biodiversity. For the breeding of livestock, it is desirable to provide a grazing plan in order to preserve natural habitats [54]. For this reason, the application of some management practices, such as adopting (1) a moderate grazing program well known as a tool of managing floristic biodiversity [55] (2) livestock’s breeding strategies in family projects well used  in arid regions in Tunisia [56] and (3) the deferred grazing, becomes a necessity for optimizing ecosystems productivity and conserving biodiversity.

Point 21: The authors sentence: (1) Association with Stipa tenacissima and Artemisia herba alba dominated the surrounding mountains (2) association with Artemisia herba alba and Haloxylon scoparium char- acterised loamy arid piles of the study area. (3) Association with Rhanterium suaveolens and Artemisia campestris defined the sandy plains. (4) Association with Thymus capitatus and Artemisia campestris marked the wadi beds.

Please change the term association in grouping, because association is a specifc syntaxon not taxon. For example Pistacia pinetum halepensis. Please change in the suggested way: i.e. Stipa tenacissima and Artemisia herba grouping or specify the reative syntaxon.

  • Thank you for pointing out this. The correction was made. Please refer page 39 Lines: 300-304.

Point 22: The fully herbarium specimens not The fully Voucher specimens

  • The correction was made Please refer page 40 line 337.

Conclusion

Point 23: Please complete with this crucial concept

After period “…sector despite its small size (100 km2) and ecological harshness”. Continue in this way:

As regard endemic or threat wild species, it is suggested to launch specific actions in order to verify if they are Crop Wild Relatives, and which is their genetic affinity with the respective parental species [Perrino and Wagensommer 2021, ].

Add these 3 references

Maxted, N.; Kell, S., Establishment of a Network for the In Situ Conservation of Crop Wild Relatives: Status and Needs. Commission on Genetic Resources for Food and Agriculture. Food and Agriculture Organization of the United Nations, Rome, Italy, 2009.

Zair, W.; Maxted, N.; Brehm, J. M.; Amri, A., Ex situ and in situ conservation gap analysis of crop wild relative diversity in the Fertile Crescent of the Middle East. Genet. Resour. Crop Evol., 2020. doi: 10.1007/s10722-020-01017-z

Perrino, E. V., Wagensommer, R. P., Crop Wild Relatives (CWR) Priority in Italy: Distribution, Ecology, In Situ and Ex Situ Conservation and Expected Actions. Sustainability, 2021, 13, 4, 1682. https://doi.org/10.3390/su13041682.

  • Thank you for this recommendation: from Line 384 to 390 we developed the suggested idea of Crop Wild Relatives and we added suggested references hopefully we have well understood your comment.

…As regard endemic, rare or threat wild species namely L. coronopifolia, A. roseum  subsp. odoratissimum and R. eriocalyx, it is suggested to launch specific actions in order to verify if they are Crop Wild Relatives (CRW), and which is their genetic affinity with the respective parental species [54]. CRW taxa are potential sources of traits beneficial to crops, such as pest or disease resistance, yield improvement or stability[66]and once identified, there is an imperative to effectively conserve these critical species in situ (i.e., in natural habitats managed as genetic reserves) and ex situ (primarily as seed in gene banks or as mature individuals in field collections) in order to underpin future world food [66,67]

Round 2

Reviewer 1 Report

Dear Authors,

I have carefully read the revised version of your manuscript  and the supplementary materials and found them improved over the previous one. However, serious gaps still persist from the point of view of the scientific nomenclature of several taxa. English has also improved, but still needs some linguistic revision. Other minor errors are also indicated in the PDF. Last but not least is the need to implement your research background in the Introduction section.

In particular:

The Introduction section should begin with a presentation of the ethnobotanical topic in general, with reference to its importance in other parts of the world. This is to give a broader, more global background before starting to deal with the local issue. I suggest to consider at least on example from other continents. After, you can deal with African and then Tunisian ethnobotanical knowledge. I recommend to start with these recent papers:
- WORLD: Salmerón-Manzano, E.; Garrido-Cardenas, J.A.; Manzano-Agugliaro, F. Worldwide Research Trends on Medicinal Plants. Int. J. Environ. Res. Public Health 2020, 17, 3376. https://doi.org/10.3390/ijerph17103376
- EUROPE: Maruca, G., Spampinato, G., Turiano, D. et al. Ethnobotanical notes about medicinal and useful plants of the Reventino Massif tradition (Calabria region, Southern Italy). Genet Resour Crop Evol 66, 1027–1040 (2019). https://doi.org/10.1007/s10722-019-00768-8
ASIA: Abdul Aziz, M.; Ullah, Z.; Pieroni, A. Wild Food Plant Gathering among Kalasha, Yidgha, Nuristani and Khowar Speakers in Chitral, NW Pakistan. Sustainability 2020, 12, 9176. https://doi.org/10.3390/su12219176
AFRICA: Omotayo, A.O.; Ndhlovu, P.T.; Tshwene, S.C.; Aremu, A.O. Utilization Pattern of Indigenous and Naturalized Plants among Some Selected Rural Households of North West Province, South Africa. Plants 2020, 9, 953. https://doi.org/10.3390/pl

Please, again: check with accuracy the authorship of the taxa mentioned in the text  and the supplementary materials. I see that often you put a dot after the name of an author: this must be putted only if the name is abbreviated, otherwise it doesn't go there.

Line 68. Sorry, maybe I didn't well understand. Have you found 165 taxa throughout the area (paragraph 2.1) of which 70 have an ethnobotanical use (paragraph 2.2)? Please, explain better. Also explain better in this paragraph why 65 ethnospecies correspond to 70 taxa (give some examples).

Lines 90-91: It is a good idea to compare more in general these data. Please, check also the following paper for your usefulness:
Gras, A.; Hidalgo, O.; D’Ambrosio, U.; Parada, M.; Garnatje, T.; Vallès, J. The Role of Botanical Families in Medicinal Ethnobotany: A Phylogenetic Perspective. Plants 2021, 10, 163. https://doi.org/10.3390/plants10010163

Lines 101-103: Their condition of "effemericity" being therophytes was also valid in the previous work cited by Le Floc'h [6]. What would be important to understand is why therophytes were used more before and not now.

Lines 146-147: There are several paper dealing with this issue. If you want, you could consider also this following, recently published in MDPI: 
- Sicari, V.; Loizzo, M.R.; Sanches Silva, A.; Romeo, R.; Spampinato, G.; Tundis, R.; Leporini, M.; Musarella, C.M. The Effect of Blanching on Phytochemical Content and Bioactivity of Hypochaeris and Hyoseris Species (Asteraceae), Vegetables Traditionally Used in Southern Italy. Foods 2021, 10, 32. https://doi.org/10.3390/foods10010032

I have noticed that scientific names often differ from those accepted on www.ipni.org and www.gbif.org. Please check them again against the nomenclatural sources you used.

Please carefully check your manuscript and the supplementary materials according to my corrections and suggestions and those of the other referees to improve your manuscript so that it can be published in the prestigious Plants journal.

Best wishes.

Author Response

The authors would like to thank the reviewers for their thoughtful comments and valuable suggestions. Please find in the attached file a point by point response to the reviewer’s comments

Reviewer 2 Report

The authors Karous et al. improved the manuscript in the correct way followed the recommendations suggested. However, some minor revisions still need to be done.

  • Please format the text, tables as suggested by the journal guidelines
  • Row 292 (page 38), the reference [54] its another, the following one. Please add and renumbered the references accordingly
  • Perrino, E. V.; Musarella, C. M.; Magazzini, P. Management of grazing Italian river buffalo to preserve habitats defined by Directive 92/43/EEC in a protected wetland area on the Mediterranean coast: Palude Frattarolo, Apulia, Italy. Euro-Mediterr. J. Environ. Integr. 2020, 6, 32. doi:10.1007/s41207-020-00235-2
  • Pay attention that at row 386 (page 42) the reference now [54], that with renumbering will become [66], its correct as reported but is different from previous. 
  • References: please add for each references doi when is available. For examples I saggested these two reference with doi but the author forgot to consideri it
  • Zair, W.; Maxted, N.; Brehm, J. M.; Amri, A., Ex situ and in situ conservation gap analysis of crop wild relative diversity in the Fertile Crescent of the Middle East. Resour. Crop Evol., 2020. doi: 10.1007/s10722-020-01017-z
  • Perrino, E. V., Wagensommer, R. P., Crop Wild Relatives (CWR) Priority in Italy: Distribution, Ecology, In Situ and Ex Situ Conservation and Expected Actions. Sustainability, 2021, 13, 1682. https://doi.org/10.3390/su13041682.

Author Response

The authors would like to thank the reviewers for their thoughtful comments and valuable suggestions. Please find  in the attached file a point by point response to the reviewer’s comments:

Round 3

Reviewer 1 Report

Dear Authors,

Your manuscript is almost ready to be accepted by Sustainability journal for its publication. All the requested improvement are indicated in attached ZIP files. Just some minor (but very important) typos and mistakes remain to be solved: all are highlighted in the attached pdf and Appendix A.

Best wishes.

Author Response

The authors would like to thank the reviewers for their thoughtful comments and valuable suggestions. We appreciate the time and effort that they have dedicated to providing their valuable feedback on my manuscript.

Please find attached   a point by  point response to your comments and concerns:
